# Detecting Invariant Manifolds in ReLU-Based RNNs

**Lukas Eisenmann**[1,2,*]**, Alena Brändle**[1,2,3,*] **Zahra Monfared**[3,4]**, and Daniel Durstewitz**[1,2,3]

{lukas.eisenmann,daniel.durstewitz}@zi-mannheim.de
[1]Dept. of Theoretical Neuroscience, Central Institute of Mental Health, Mannheim, Germany
[2]Faculty of Physics and Astronomy, Heidelberg University, Heidelberg, Germany
[3]Interdisciplinary Center for Scientific Computing, Heidelberg University, Germany
[4]Faculty of Mathematics and Computer Science, Heidelberg University, Heidelberg, Germany
[*]These authors contributed equally

## Abstract

Recurrent Neural Networks (RNNs) have found widespread applications in machine learning for time series prediction and dynamical systems reconstruction, and experienced a recent renaissance with improved training algorithms and architectural designs. Understanding why and how trained RNNs produce their behavior is important for scientific and medical applications, and explainable AI more generally. An RNN's dynamical repertoire depends on the topological and geometrical properties of its state space. Stable and unstable manifolds of periodic points play a particularly important role: They dissect a dynamical system's state space into different basins of attraction, and their intersections lead to chaotic dynamics with fractal geometry. Here we introduce a novel algorithm for detecting these manifolds, with a focus on piecewise-linear RNNs (PLRNNs) employing rectified linear units (ReLUs) as their activation function. We demonstrate how the algorithm can be used to trace the boundaries between different basins of attraction, and hence to characterize multistability, a computationally important property. We further show its utility in finding so-called homoclinic points, the intersections between stable and unstable manifolds, and thus establish the existence of chaos in PLRNNs. Finally we show for an empirical example, electrophysiological recordings from a cortical neuron, how insights into the underlying dynamics could be gained through our method.

## 1 Introduction

Recurrent neural networks (RNNs) are widely employed for time series forecasting (Park et al., 2018; Gu and Dao, 2023) and dynamical systems (DS) reconstruction (Hess et al., 2023; Brenner et al., 2022; 2024b; Platt et al., 2023), especially in scientific applications like climate modeling (Patel and Ott, 2023) or neuroscience (Durstewitz et al., 2023), as well as in medical domains. RNNs experienced a recent revival due to the advance of new powerful training algorithms that avoid vanishing or exploding gradients (Hess et al., 2023), and novel network architectures (Schmidt et al., 2021; Rusch and Mishra, 2021; Peng et al., 2023; Gu and Dao, 2023; Rusch et al., 2022), in particular in the context of state space models (which are essentially linear RNNs with nonlinear readouts and input gating; Gu and Dao (2023); Orvieto et al. (2023)). What lags behind, like in many other areas of deep learning, is a thorough theoretical understanding of the behavior of these systems and how they achieve the tasks they were trained on. Yet, such an understanding is crucial especially in scientific and medical areas where we are often interested in using trained RNNs as surrogate models for the underlying DS, providing mechanistic insight into the dynamical processes that generated the observed time series.

Formally, RNNs are recursive maps, hence discrete-time dynamical systems (Mikhaeil et al., 2022). Dynamical systems theory (DST) offers a rich repertoire of mathematical tools for analyzing the behavior of such systems (Guckenheimer and Holmes, 2013). Yet, exploration of high-dimensional

RNN dynamics remains challenging due to limitations in current numerical methods, which struggle to scale and often only yield approximate results (Katz and Reggia, 2018; Golub and Sussillo, 2018).

The dynamical behavior of a system is governed by its state space topology and geometry (Hasselblatt and Katok, 2002), most prominently topological objects like attractors, such as stable fixed points, cycles, or chaotic sets, which determine the system's long-term behavior. Similarly important are the stable and unstable manifolds of fixed points and cycles, although they received much less attention in the scientific and ML communities. Stable manifolds are the sets of points in a system's state space that converge towards an equilibrium or periodic orbit in forward-time, while unstable manifolds, conversely, are the sets of points that converge to it in backward-time. Stable manifolds of saddle points delineate the boundaries between basins of attraction in multistable systems, that is, systems that harbor multiple attractor objects between which it can be driven back and forth by perturbations like external inputs or noise (Feudel et al., 2018). Multistability has been hypothesized to be an important computational property of RNNs, most prominently in computational neuroscience where it has been linked to working memory (Durstewitz et al., 2000) or decision making (Wang, 2008).

Tracing out stable and unstable manifolds is also important for finding homo- and heteroclinic orbits, which connect a cyclic point to itself or to another such point, respectively. These orbits provide a skeleton for the dynamics, forming structures like separatrix cycles (Perko, 2001) and heteroclinic channels (Rabinovich et al., 2008) which have been implied in flexible sequence generation. Intersections between stable and unstable manifolds further give rise to so-called homoclinic points which create sensitive regions in a system's state space associated with chaos (Wiggins, 1988). Chaotic behavior, in turn, or regimes at the edge-of-chaos, have been associated with increased expressivity (larger function classes that can be emulated) and computational power in RNNs (Siegelmann and Sontag, 1992; Bertschinger and Natschläger, 2004; Pereira-Obilinovic et al., 2023) (Mikhaeil et al., 2022; Hess et al., 2023).

Here we introduce a novel algorithm for computing the stable and unstable manifolds of fixed and cyclic points for the class of piecewise-linear (PL) RNNs (PLRNNs), which use PL nonlinearities like the ReLU as their activation function. Unlike traditional techniques designed for smooth dynamical systems—such as numerical continuation methods (Krauskopf et al., 2007) —the proposed method explicitly exploits the piecewise-linear structure of PLRNNs to enable the exact location of stable and unstable manifolds. In contrast to ODE systems, methods for discrete-time systems are much more scarce, especially if these involve discontinuities in their Jacobians like ReLU-based RNNs. PLRNNs have emerged as one of the most powerful classes of models for DS reconstruction (Hess et al., 2023; Brenner et al., 2022; 2024b; Nassar et al., 2019), partly because the linear subspaces of such models support the indefinite online retention of memory contents without exploding or vanishing gradients (Schmidt et al., 2021; Orvieto et al., 2023; Gu and Dao, 2023), and partly because their PL structure makes them more tractable (Coombes et al., 2024). In fact, PL systems have been popular in engineering (Bemporad et al., 2000; Carmona et al., 2002; Storace and De Feo, 2004) and mathematics (Alligood et al., 1996; Avrutin et al., 2019; Coombes et al., 2024; Simpson, 2023) for decades for these reasons, and a variety of PL models similar in structure to PLRNNs exist, like threshold-linear networks (TLNs; Curto and Morrison (2023); Parmelee et al. (2022); Morrison et al. (2024); Hahnloser and Seung (2000)), switching linear DS (SLDS; Linderman et al. (2016; 2017)), or Lur'e systems in control theory (Su et al., 2023; Waitman et al., 2017; Bill et al., 2016), with a substantial body of mathematical theory built around them (Parmelee et al., 2022; Morrison et al., 2024; Brenner et al., 2024a; Avrutin et al., 2019; Amaral et al., 2006; Simpson, 2023; Coombes et al., 2024; Casdagli, 1989; Ives and Dakos, 2012; Costa et al., 2019). Further, efficient algorithms for exactly localizing fixed and cyclic points in polynomial time exist (Eisenmann et al., 2023), on which we will build here. We illustrate how our algorithm can be employed to delineate the boundaries of basins of attraction and to detect homoclinic intersections, thus establishing the existence of chaos, and show how it can be used on empirical data to gain insight into dynamical mechanisms.

## 2 RELATED WORK

While there is an extensive literature by now on DS reconstruction with RNNs (Brenner et al., 2022; 2024a; Hess et al., 2023; Patel and Ott, 2023; Platt et al., 2023; Cestnik and Abel, 2019), work on algorithms for dissecting topological properties of trained systems is much more scarce. Existing research is almost exclusively focused on algorithms for locating stable and unstable fixed points

and cycles (Golub and Sussillo, 2018; Eisenmann et al., 2023), but not the invariant manifolds associated with them, despite their crucial role in structuring the state space and dynamics. More generally in the DST literature, a related class of methods are so-called continuation methods which numerically trace back important curves and manifolds in dynamical systems, mostly for systems of ordinary differential equations (Krauskopf et al., 2007; Osinga, 2014). Recently methods have been developed specifically for PL maps (Simpson, 2023), but, similar to continuation methods, they seriously suffer from the curse of dimensionality. Thus, current methods effectively work only for very low-dimensional ($\leq 5d$) systems, in contrast to the size of many modern RNNs used for time series forecasting or dynamical systems reconstruction. The specific structure of PL maps, of which all types of PLRNNs are specific examples, considerably eases the computation of certain topological properties. This also provided the basis for the SCYFI algorithm for locating fixed points and cycles, which scales polynomially, in fact often linearly, with the dimensionality of the PLRNN's latent space (Eisenmann et al., 2023). To our knowledge, no existing method efficiently detects stable and unstable manifolds in discrete-time RNNs. Our approach fills this gap by leveraging the PL nature of ReLU-based RNNs to construct manifolds directly, bypassing the limitations of traditional techniques.

## 3 METHODS

### 3.1 PLRNN ARCHITECTURES

A PLRNN (Durstewitz, 2017) is simply a ReLU-based RNN

$$\boldsymbol{z}_t = \boldsymbol{A}\boldsymbol{z}_{t-1} + \boldsymbol{W}\Phi(\boldsymbol{z}_{t-1}) + \boldsymbol{h}, \tag{1}$$

where $\boldsymbol{A} \in \mathbb{R}^{M \times M}$ is a diagonal matrix, $\boldsymbol{W} \in \mathbb{R}^{M \times M}$ an off-diagonal matrix, the element-wise ReLU non-linearity $\Phi(\cdot) = \max(0, \cdot)$, and $\boldsymbol{h} \in \mathbb{R}^M$ a bias term. Several variants of the PLRNN have been introduced to enhance its expressivity or reduce its dimensionality (Brenner et al., 2022), of which we picked for demonstration here the shallow PLRNN (shPLRNN; (Hess et al., 2023)), $\boldsymbol{z}_t = \boldsymbol{A}\boldsymbol{z}_{t-1} + \boldsymbol{W}_1\Phi(\boldsymbol{W}_2\boldsymbol{z}_{t-1} + \boldsymbol{h}_2) + \boldsymbol{h}_1$ with $\boldsymbol{W}_1 \in \mathbb{R}^{M \times H}$, $\boldsymbol{W}_2 \in \mathbb{R}^{H \times M}$, and the recently proposed almost-linear RNN (ALRNN; (Brenner et al., 2024a)), which attempts to use as few nonlinearities as needed for the problem at hand, $\Phi(\boldsymbol{z}_t) = [\boldsymbol{z}_{1,t}, \dots, \boldsymbol{z}_{M-P,t}, \max(0, \boldsymbol{z}_{M-P+1,t}), \dots, \max(0, \boldsymbol{z}_{M,t})]$.

The PL structure of these PLRNNs can be exposed by rewriting the ReLU in Equation (1) as

$$\boldsymbol{z}_t = F_\theta(\boldsymbol{z}_{t-1}) = (\boldsymbol{A} + \boldsymbol{W}\boldsymbol{D}_{t-1})\boldsymbol{z}_{t-1} + \boldsymbol{h}, \tag{2}$$

$\boldsymbol{D_t} := \operatorname{diag}(\boldsymbol{d_t})$ with $\boldsymbol{d_t} = (d_{1,t}, d_{2,t}, \dots, d_{M,t})$ and $d_{i,t} = 0$ if $z_{i,t} \leq 0$ and $d_{i,t} = 1$ otherwise (Monfared and Durstewitz, 2020). For the ALRNN, we have $d_{1:M-P,t} = 1 \forall t$. Hence, we have $2^M$ linear subregions for the standard PLRNN, $\leq \sum_{k=0}^{M} \binom{H}{k}$ for the shPLRNN (Pals et al., 2024) , and $2^P$ for the ALRNN.

### 3.2 MATHEMATICAL PRELIMINARIES

Recall that a fixed point of a recursive map $\boldsymbol{x}_t = F(\boldsymbol{x}_{t-1})$ is a point $\boldsymbol{x}^*$ for which $\boldsymbol{x}^* = F(\boldsymbol{x}^*)$, and a cyclic point with period $m$ is a fixed point of the $m$-times iterated map, i.e. such that $\boldsymbol{x}_{t+m} = F^m(\boldsymbol{x}_t) = \boldsymbol{x}_t$ (hence, a fixed point is a period-1 point). An $m$-cycle is a periodic sequence $\{\boldsymbol{x}_1^* \dots \boldsymbol{x}_m^*\}$ of such points with all points distinct, $\boldsymbol{x}_i^* \neq \boldsymbol{x}_j^* \ \forall 1 \leq i < j \leq m$.

**Definition 1** (Un-/stable manifold). Let $F : \mathbb{R}^M \to \mathbb{R}^M$ be a map and $\boldsymbol{p}$ be a hyperbolic period-$m$ cyclic point of $F$. The *local stable manifold* of $\boldsymbol{p}$, $W_{\text{loc}}^{\{+1\}}(\boldsymbol{p})$, is defined as

$$W_{\text{loc}}^{\{+1\}}(\boldsymbol{p}) := \{\boldsymbol{x} \in \mathbb{R}^M \ : \ F^{nm}(\boldsymbol{x}) \to \boldsymbol{p} \quad \text{as} \quad n \to \infty\}.$$

The *local unstable manifold* of $\boldsymbol{p}$, $W_{\text{loc}}^{\{-1\}}(\boldsymbol{p})$, is defined as

$$W_{\text{loc}}^{\{-1\}}(\boldsymbol{p}) := \{\boldsymbol{x} \in \mathbb{R}^M \ : \ F^{-nm}(\boldsymbol{x}) \to \boldsymbol{p} \quad \text{as} \quad n \to \infty\}.$$

The *global stable manifold* is the union of all preimages of the local stable manifold, and the *global unstable manifold* is created by the union of all images (forward iterations) of the local unstable manifold (Patra and Banerjee, 2018):

$$W^{\{+1\}}(\boldsymbol{p}) := \bigcup_{n=1}^{\infty} F^{-n}\left(W_{\text{loc}}^{\{+1\}}(\boldsymbol{p})\right), \qquad W^{\{-1\}}(\boldsymbol{p}) := \bigcup_{n=1}^{\infty} F^n\left(W_{\text{loc}}^{\{-1\}}(\boldsymbol{p})\right). \tag{3}$$

If the map $F$ is noninvertible (i.e., does not have a unique inverse), the (global) stable manifold could be disconnected, making its computation hard as we need to trace back disconnected sets of points in time to determine it. However, most commonly we aim to approximate smooth continuous-time DS $\dot{\boldsymbol{x}} = f(\boldsymbol{x})$ in DS reconstruction through our RNN map $F_\theta$. For such systems, the Picard-Lindelöf theorem guarantees uniqueness of solutions and the flow (solution) operator $\phi(t, \boldsymbol{x}_0) = \boldsymbol{x}_0 + \int_0^t f(\boldsymbol{u}) \, \mathrm{d}\boldsymbol{u}$ will be a diffeomorphism (Perko, 2001), hence invertible, such that it is reasonable to assume (or enforce) invertibility for $F_\theta$ as well.

For a PLRNN, since it is a PL map, in each linear subregion we have the following explicit (non-recursive) expression for the dynamics along trajectories ($\boldsymbol{z}_t = \tilde{\boldsymbol{W}} \boldsymbol{z}_{t-1} + \boldsymbol{h}, \tilde{\boldsymbol{W}} := \boldsymbol{A} + \boldsymbol{W} \boldsymbol{D}(\boldsymbol{z}_{t-1})$):

$$\boldsymbol{z}_t = \sum_{\substack{j=1 \\ j \neq i}}^n c_j \lambda_j^t \boldsymbol{v}_j \; + \; \lambda_i^t \sum_{r=1}^d c_r \frac{t^{r-1}}{(r-1)!} \boldsymbol{w}_r \; + \; \sum_{k=0}^{t-1} \tilde{\boldsymbol{W}}^k \boldsymbol{h}. \tag{4}$$

where the $c_i \in \mathbb{R}$ are determined from the initial condition, $\boldsymbol{v}_i$ are the eigenvectors, $\boldsymbol{w}_i$ are generalized eigenvectors (associated with defective eigenvalues with geometric multiplicity less than the algebraic). This formula describes the exponential evolution for all non-defective eigenvalues $\lambda_j$ and polynomially modulated exponentials for the defective eigenvalue $\lambda_i$, see Appx. B for details (Perko, 2001; Hirsch et al., 2013). While this expression is for single orbits, it describes how curvature in the manifolds can arise as we cross the boundaries of linear subregions (cf. sect. 3.3).

Stable manifolds of saddle objects segregate the state space into different basins of attraction (Ganatra and Banerjee, 2022), which are sets of points from which the state evolves toward one or the other attractor. Formally, they are given by (Alligood et al., 1996):

**Definition 2** (**Basin of attraction**). Let $F : \mathbb{R}^M \to \mathbb{R}^M$ be a map. The *basin of attraction* of an attractor $\mathcal{A}$ is the largest open set $B(\mathcal{A}) \subseteq \mathbb{R}^M$ (containing $\mathcal{A}$) such that for every point $\boldsymbol{x} \in B(\mathcal{A})$, the iterates of $\boldsymbol{x}$ under the map $F$ converge to $\mathcal{A}$ in the forward limit:

$$B(\mathcal{A}) \; = \; \left\{ \boldsymbol{x} \in \mathbb{R}^M : d(F^k(\boldsymbol{x}), \mathcal{A}) \to 0 \text{ as } k \to \infty \right\},$$

where $F^k$ denotes the $k$-times iterated map $F$.

Stable and unstable manifolds are crucially important for the system dynamics not only because they dissect the state space into different regions of flow, but also because their intersections can give rise to complex types of dynamics like heteroclinic channels or chaos (Rabinovich et al., 2008; Perko, 2001).

**Definition 3** (**Homoclinic orbit**). Let $\boldsymbol{p}$ be a saddle fixed point (or saddle cycle) of the map $F :$ $\mathbb{R}^M \to \mathbb{R}^M$. A *homoclinic orbit* is a trajectory $\mathcal{O}_{hom} = \{\boldsymbol{x}_n\}_{n \in \mathbb{N}}$ that connects $\boldsymbol{p}$ to itself, i.e., $\boldsymbol{x}_n \to \boldsymbol{p}$ as $n \to \pm\infty$. Hence,

$$\mathcal{O}_{hom} \subset W^{\{+1\}}(\boldsymbol{p}) \; \cap \; W^{\{-1\}}(\boldsymbol{p}).$$

If $\boldsymbol{x} \neq \boldsymbol{p}$ is a point where the stable and unstable manifolds of $\boldsymbol{p}$ intersect, then $\boldsymbol{x}$ is referred to as a homoclinic point or intersection (Patra and Banerjee, 2018; Perko, 2001). Such an intersection of the stable and unstable manifolds leads to a horseshoe structure associated with a fractal geometry and chaos (see Appx. I, Wiggins (1988)).

**Definition 4** (**Heteroclinic orbit**). Let $\boldsymbol{p}$ and $\boldsymbol{q}$ be two *distinct* saddle fixed points (or saddle cycles) of the map $F$. A *heteroclinic orbit* $\mathcal{O}_{het} = \{\boldsymbol{x}_n\}_{n \in \mathbb{N}}$ connects two different fixed points $\boldsymbol{p}$ and $\boldsymbol{q}$, that is, $\boldsymbol{x}_n \to \boldsymbol{p}$ as $n \to +\infty$ and $\boldsymbol{x}_n \to \boldsymbol{q}$ as $n \to -\infty$. Heteroclinic intersections (or points) can be defined similarly to homoclinic intersections, and, like homoclinic points, inevitably lead to chaos (Wiggins, 1988; Patra and Banerjee, 2018).

### 3.3 LOCATING UN-/STABLE MANIFOLDS

For our manifold detection algorithm, we exploit the PL structure of PLRNNs, which allows for an analytical construction of the manifolds within each linear subregion. We further assume that $F_\theta$ is invertible (see above), such that the global un-/stable manifolds will be connected sets. We start by using a previously proposed algorithm, SCYFI (Eisenmann et al. (2023), see also Pals et al. (2024)), to locate saddle (cyclic) points $\boldsymbol{p}$ of interest, for which we would like to construct the un-/stable manifolds. We then proceed as follows (Algo. 1; see Fig. 1 for illustration of these steps in $1d$):

---

**Algorithm 1** Manifold Construction

---

**Hyperparameters:** $N_s$: Number of support points to be sampled
$\qquad\qquad\quad$ $N_{\text{iter}}$: Maximum number of iterations
**Input:** $\boldsymbol{p}$ : Periodic point
$\qquad$ $\sigma \in \{-1, 1\}$: flag indicating whether stable $(+1)$ or unstable $(-1)$ manifold is to be computed
$\qquad$ $\boldsymbol{\theta}$: PLRNN parameters
**Output:** $\{Q^\sigma\}$: Support & eigenvectors parameterizing the manifold in each subregion

1: $E^\sigma \leftarrow \text{GETEIGENVEC}(\boldsymbol{\theta}, \boldsymbol{D}(\boldsymbol{p}))$ $\qquad\qquad\qquad\qquad$ ▷ Compute eigenvectors at $\boldsymbol{p}$
2: $Q^\sigma \leftarrow \{\{\boldsymbol{p}, E^\sigma\}\}$ $\qquad\qquad\qquad\qquad$ ▷ Initialize set of manifold specifications
3: $K_{\text{seed}}, K_{\text{visited}} \leftarrow \{\text{GETSUBREGIONID}(\boldsymbol{p})\}$ $\qquad$ ▷ Initialize set of seed & visisted subregions
4: **while** $n \leq N_{\text{iter}} \wedge \neg\text{isempty}(K_{\text{seed}})$ **do**
5: $\quad$ **for** $k \in K_{\text{seed}}$ **do** $\qquad\qquad\qquad\qquad\qquad$ ▷ Loop over regions to expand from
6: $\qquad$ $S^\sigma \leftarrow \text{SAMPLEPOINTS}(Q_k^\sigma, N_s)$ $\qquad\qquad$ ▷ Sample $N_s$ points on the manifold
7: $\qquad$ $\tilde{S}^\sigma \leftarrow \text{PROPAGATETONEXTREGION}(S^\sigma, \boldsymbol{\theta})$ ▷ Propagate forward/backward using $F_{\boldsymbol{\theta}}^\sigma$
8: $\qquad$ $K_{\text{con}(k)} \leftarrow \text{GETSUBREGIONID}(\tilde{S}^\sigma) \setminus K_{\text{visited}}$ $\qquad$ ▷ Connecting regions not yet visited
9: $\qquad$ **for** $j \in K_{\text{con}(k)}$ **do** $\qquad\qquad\qquad\qquad$ ▷ Loop over connecting regions
10: $\qquad\quad$ $\lambda_j \leftarrow \text{GETEIGENVAL}(\boldsymbol{\theta}, \boldsymbol{D}_j)$ $\qquad\qquad\qquad$ ▷ Compute eigenvalues
11: $\qquad\quad$ **if** $\text{Im}(\lambda_j) \neq 0 \vee (\text{multiplicity}_{\text{geo}}(\lambda_j) < \text{multiplicity}_{\text{alg}}(\lambda_j))$ **then**
12: $\qquad\qquad$ $Q^\sigma \leftarrow Q^\sigma \cup \{\text{KPCA}(\tilde{S}_j^\sigma), (\tilde{S}_j^\sigma)_1\}$ $\qquad$ ▷ Determine manifold by kernel-PCA
13: $\qquad\quad$ **else**
14: $\qquad\qquad$ $Q^\sigma \leftarrow Q^\sigma \cup \{\text{PCA}(\tilde{S}_j^\sigma), (\tilde{S}_j^\sigma)_1\}$ $\qquad\qquad$ ▷ Determine manifold by PCA
15: $\qquad\quad$ **end if**
16: $\qquad$ **end for**
17: $\qquad$ $K_{\text{visited}} \leftarrow K_{\text{visited}} \cup K_{\text{con}(k)}$ $\qquad\qquad\qquad$ ▷ Update set of visited regions
18: $\quad$ **end for**
19: $\quad$ $K_{\text{seed}} \leftarrow K_{\text{con}}$ $\qquad\qquad\qquad\qquad$ ▷ Make all connecting regions new seed
20: **end while**
21: **return** $\{Q^\sigma\}$ $\qquad\qquad\qquad\qquad\qquad$ ▷ Manifold segments in all subregions

---

1. *Locally*, in the linear subregion harboring $\boldsymbol{p}$, the unstable (resp. stable) manifold of $\boldsymbol{p}$ is simply given by the affine subspaces spanned by the eigenvectors with absolute eigenvalues within (stable) or outside of (unstable) the unit circle, and hence can easily be computed in closed form from the local Jacobian $\boldsymbol{A} + \boldsymbol{W}\boldsymbol{D}(\boldsymbol{p})$. Note this remains true for complex eigenvalues (associated with spiral points), as these always come as conjugate pairs and hence always span a planar subspace.

2. To extend the un-/stable manifolds across the boundaries into neighboring linear regions, we randomly sample $N_s$ points on the manifold and propagate them forward by $F_\theta$ (unstable) or backward by $F_\theta^{-1}$ (stable) into neighboring regions. The hyper-parameter $N_s$ is chosen such that with high likelihood all neighboring subregions into which the manifold extends are reached, which is mostly fulfilled if we choose $N_s = k_{\text{neigh}} \times d$, the total number of neighboring subregions times the manifold dimension.[1] For the inversion of $F_\theta$, specifically, we need to solve $\boldsymbol{z}_{t-1} = (\boldsymbol{A} + \boldsymbol{W}\boldsymbol{D}_{t-1})^{-1}(\boldsymbol{z}_t - \boldsymbol{h})$. A solution to this eq. needs to be self-consistent, i.e. the signs of the $z_{i,t-1}$ on the l.h.s. need to be consistent with the entries $d_{i,t-1}$ in $\boldsymbol{D}_{t-1}$ on the r.h.s. To address this, we introduce a simple heuristic: 1) Perform a backward step using the current linear subregion $\boldsymbol{D}_t$; 2) perform a forward step using the resulting candidate solution $\boldsymbol{z}_{t-1}^*$; 3) if $\boldsymbol{z}_t == F_\theta(\boldsymbol{z}_{t-1}^*)$, we are done; 4) if not, we take the candidate's linear subregion $\boldsymbol{D}(\boldsymbol{z}_{t-1}^*)$ to attempt a new inversion; 5) if this fails, we start checking the neighboring regions by iteratively flipping bits in $\boldsymbol{D}_{t-1}$ (see Appx. E, Algorithm 2, for details). The efficiency of this procedure comes from the fact that usually the candidate will lie in the correct linear subregion, and – if this is not the case – it usually crosses only into neighboring regions.

3. The manifolds cannot change their dimensionality across subregions, *but they may become curved along one or more directions as they cross into a neighboring subregion* (for instance, if the dynamics in the new subregion is determined by a spiral, see example in Fig. 11). In theory, we only need to compute as many points in each subregion as required to uniquely anchor the manifold segment within that region. As we know the type of curvature qua Eq. 4, ideally these

---

[1]If unsure, one could increase $N_s$ stepwise until the number of subregions reached plateaus.

would just be $d + 1$ points for a $d < M$-dimensional manifold even in the curved case, see Appx. D. In practice, numerical issues (see below) and the type of approximation used may require sampling more points. If the manifold segment is non-curved, we use PCA as a quick and simple means to obtain the affine subspace from $d + 1$ support points. If the manifold is curved, we use kernel-PCA (see Appx.C). The $\geq d$ eigenvectors with non-zero eigenvalues, together with at least one support point, exactly define manifold segment $Q_k^\sigma$ in subregion $k$ (see Appx. D for another type of approximation for the manifold segment in each subregion).

4. We then iterate steps 1-3 for at most $N_{iter}$ times, recursively expanding the manifolds across the whole state space. The algorithm's final output will be for all $K$ subregions the set of all eigenvectors with at least one support vector positioning each manifold segment.

Note that within each linear region the construction of the manifold is thus analytical from the $\geq d + 1$ support points, while across regions we recursively assemble the whole manifold following Eq. 3. In systems with highly disparate timescales, however, numerical issues may arise as sampled support points mainly spread along the fast eigendirections with little variation along the slower eigendirections, leading to poor numerical approximation of a manifold. To balance the contributions from different directions we therefore adjust sampling density inversely to eigenvalue magnitude, with an additional hyperparameter $c$ controlling this weighting (see Appx.C).

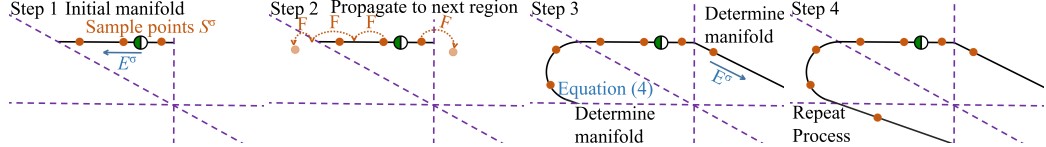

Figure 1: One-dimensional illustration of the iterative procedure for computing stable manifolds with subregion boundaries indicated in purple-dashed. Step 1: The stable manifold (black) is initialized using the stable eigenvector $E^{\{+1\}}$ (blue) of the saddle point (green), and sample points (orange) are placed along it. Step 2: These points are propagated until they enter another linear subregion. Step 3: A new segment of the manifold is determined. Step 4: Repeating this process iteratively reconstructs the full global structure of the stable manifold. For a trained model example see Fig. 8.

### 3.4 ENFORCING MAP INVERTIBILITY BY REGULARIZATION

In designing our algorithm, we relied on invertibility of the RNN map $F_\theta$ in the sense that $\exists! \mathbf{D}_{t-1}$ for which $F_\theta\left(F_\theta^{-1}(\mathbf{z}_t, \mathbf{D}_{t-1})\right) = \mathbf{z}_t$. This is a reasonable assumption as the flow map $\phi(t, \boldsymbol{x}_0)$ for most underlying ODE systems of interest, which we attempt to approximate, is invertible due to Picard-Lindelöf (Perko, 2001). However, empirically, invertibility of $F_\theta$ is not always guaranteed (depending on the quality of the approximation). Thus, we enforce this condition by regularization during RNN training (for training itself we used the previously proposed shPLRNN (Hess et al., 2023) or ALRNN (Brenner et al., 2024a), see sect. 3.1). $F_\theta$ is invertible, if the determinants of the Jacobian matrices of neighboring subregions have the same sign (sign condition) (Fujisawa et al., 1972). This can be enforced (for any type of ReLU-based RNN) by adding the following regularization term to the RNN training loss (commonly the MSE loss, see Appx. E):

$$\mathcal{L}_{\text{reg}} = \lambda \cdot \frac{1}{|\mathcal{S}_{\text{reg}}|} \sum_{i \in \mathcal{S}_{\text{reg}}} \max\left(0, -\det(\boldsymbol{J}_i)\right), \tag{5}$$

computed across a small subset $\mathcal{S}_{\text{reg}}$ of linear subregions, where $\boldsymbol{J}_i = \boldsymbol{A} + \boldsymbol{W}\boldsymbol{D}_i$ is the Jacobian in subregion $i$, and $\lambda$ a regularization parameter. As shown in Fig. 2A (left), strategic sampling of only 1% of the linear subregions $\mathcal{S}$ (which are traversed by actual trajectories (cf. (Brenner et al., 2024a))) is sufficient to ensure almost full invertibility of $F_\theta$ across the whole state space. At the same time, it hardly affects runtime (Fig. 2A, top-right) and reconstruction performance (Fig. 2A, bottom). Since invertible flows are an inherent property of many, if not most, DS of scientific interest, we would furthermore expect that this regularization does not hamper, or even improves, the reconstruction of DS from data, in particular if the systems carry an intrinsic time reversibility (Huh et al., 2020). This is confirmed in Fig. 2B-C which shows that with the invertibility regularization in place, training of a PLRNN on a 10d damped nonlinear oscillator converges significantly faster to a good solution

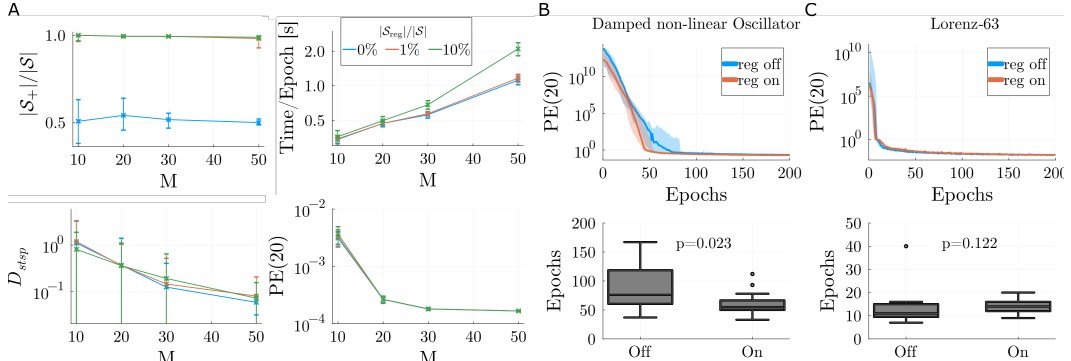

Figure 2: A) Top-left: Relative proportion of subregions with a positive determinant of the Jacobian ($\det(\boldsymbol{J}) > 0$), $\mathcal{S}_+$, as a function of latent space dimensionality $M$ for different proportions $|\mathcal{S}_{\mathrm{reg}}|/|\mathcal{S}|$. Medians $\pm$ interquartile range are shown. Top-right: Runtime and reconstruction quality (bottom) as a function of latent space dimensionality $M$ for different proportions $|\mathcal{S}_{\mathrm{reg}}|/|\mathcal{S}|$ of subregions for which invertibility was enforced by regularization ($\lambda = 0.1\exp(M)$ in Eq. 5). Means across 100 different training runs on the Lorenz-63 system $\pm$ SD are shown. Reconstruction quality was assessed through (dis-)agreement in attractor geometry (bottom-left; $D_{\mathrm{stsp}}$, see Appx. F) and 20-step-ahead prediction error (bottom-right). B) Top: 20-step-ahead prediction error, PE(20), as a function of the number of training epochs when the invertibility regularization, Eq. 5, was turned off (blue) vs. on (orange), for a damped nonlinear oscillator. Bottom: Convergence to a predefined performance criterion (PE(20) $\leq 0.5$) was significantly faster with the regularization turned on vs. off. Median across 20 trained models, error bands = interquartile range. C) Same as B for Lorenz-63 system.

($p = 0.023$, Mann-Whitney U-test) than without the regularization (Fig. 2B), while hardly affecting performance on other systems like the chaotic Lorenz-63 (Lorenz, 1963) system (Fig. 2C; $p = 0.122$).

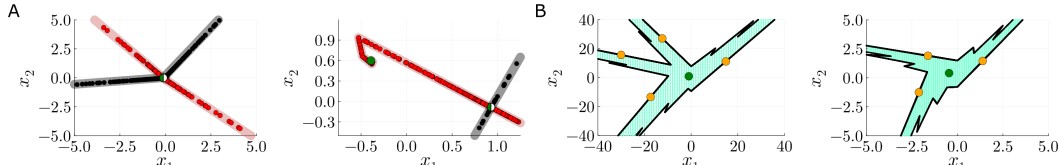

Figure 3: Model validation. A) Two examples of saddle points (half-green) with stable (gray solid lines) and unstable (red solid lines) manifolds determined by our algorithm, and points (black/ red dots, respectively) sampled by the analytical resp. backward/ forward map, showing that these all fall onto the analytically determined manifolds. B) Basins of attraction (light green, confirmed by sampling initial conditions and tracing their trajectories) of a stable fixed point (green dot) delineated by the stable manifold (black) of a 4-cycle (left) or 3-cycle (right).

## 4 DELINEATING BASINS OF ATTRACTION

**Basic methods validation** We first validate our algorithm on a simple toy example, a two-dimensional PL map for which we have analytical forms for the inverse and the fixed points (see Appx. H.1 for details). We chose parameters to produce a simple test case with only a single saddle point. Fig. 3A confirms that points produced by backward (resp. forward) iterating the map all lie on the stable (resp. unstable) manifold as determined by our algorithm. In this simple 2d example, the manifolds correspond to line segments. Changing the PL map's parameters slightly, we obtain a stable fixed point coexisting with a saddle period-4 (Fig. 3B, left) or period-3 (Fig. 3B, right) cyclic point (Gardini et al., 2009). Tracing back the stable manifold of the period-4 or period-3 saddle using our algorithm, we obtain the boundaries of the basins of attraction of the resp. fixed point (Fig. 3B). Fig. 3B further confirms this solution for the basin perfectly agrees with the 'classical' (not

scalable) numerical approach of drawing initial conditions on a grid in state space, and observing their behavior in the limit $t \to \infty$.

In these simple $2d$ test cases the quality of the manifold reconstruction can easily be verified visually. In higher dimensions, however, this will no longer be possible, and so we also introduce a simple metric to quantify the reconstruction quality. For a point $\boldsymbol{x}_0 \in \mathcal{U}$ sampled randomly either from a neighborhood $\mathcal{U}$ containing the relevant data range (see Appx. G for details), or from the reconstructed manifold of a saddle point $\boldsymbol{p}$, $\boldsymbol{x}_0 \in W^\sigma(\boldsymbol{p}) \cap \mathcal{U}$, with $\sigma \in \{-1, +1\}$ indicating whether we are considering the unstable or stable manifold, we compute

$$\delta_\sigma(\boldsymbol{x}_0) := \frac{\min_{k\sigma \geq 0} \|F_{\boldsymbol{\theta}}^k(\boldsymbol{x}_0) - \boldsymbol{p}\|_2^2}{\|\boldsymbol{x}_0 - \boldsymbol{p}\|_2^2} \in [0, 1], \tag{6}$$

For points randomly sampled from $\mathcal{U}$ or from $W^\sigma(\boldsymbol{p}) \cap \mathcal{U}$ for the example above, Fig.6A in Appx.G shows the distribution of $\delta_\sigma$, confirming $\delta_\sigma \approx 0$ for points sampled on the manifolds, while off the manifold $\delta_\sigma$ naturally spreads across a broader range. For the latter we assign an index $I_\mathcal{U}[\delta_\sigma(\boldsymbol{x}_0 \in \mathcal{U}) > \delta_\sigma^{\max}] \in \{0, 1\}$ with $\delta_\sigma^{\max} := \max[\delta_\sigma(\boldsymbol{x}_0 \in W^\sigma(\boldsymbol{p}) \cap \mathcal{U})]$. In the following, we report $\Delta_\sigma := \langle I_\mathcal{U} \rangle - \tilde{\delta}_\sigma(\boldsymbol{x}_0 \in W^\sigma(\boldsymbol{p}) \cap \mathcal{U})$ as quality statistic, where $\langle \cdot \rangle$ is the mean and $\tilde{\delta}_\sigma$ the median.

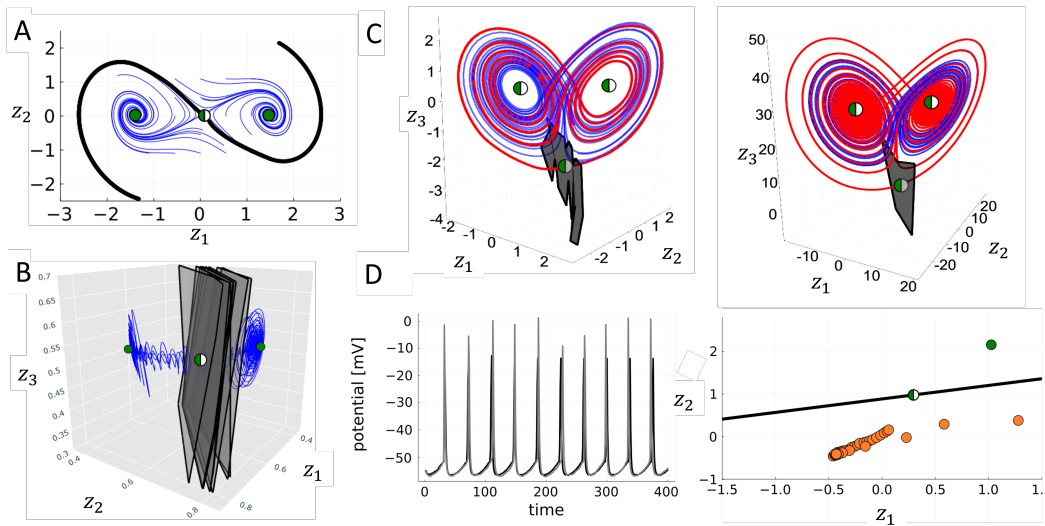

Figure 4: A) Reconstruction of the Duffing system (shPLRNN, $M = 2$, $H = 10$). Ground truth trajectories in blue, identified fixed points in green, and in gray the stable manifold of the saddle separating the two basins of attraction as determined by algorithm 1. B) 3d subspace of the state space of an ALRNN ($M = 15$, $P = 6$) trained on a 2-choice decision making task. Two point attractors (green) were identified, with the stable manifold (black/gray) of a saddle (half-green) in the center separating the basins. The basin boundary consists of different segments, projected down from a truly 15d space into a 3d subspace (accounting for some of the 'folded' appearance) and approximated linearly for visualization. C) Reconstruction for a shPLRNN ($M = 3$, $H = 20$) trained on the Lorenz63 system. The system has two saddle-spirals in the center of the two lobes and a saddle at the bottom. In black and red are the stable and unstable manifolds, respectively, of the saddle as identified by our algorithm (left), while on the right as computed by numerical continuation of the *original* system. The close agreement indicates the shPLRNN has correctly recovered the state space structure, although having been trained on data from the actual attractor only. D) ALRNN ($M = 25$, $P = 6$) trained on electrophysiological recordings. Left: Time series of membrane voltage (true: gray, simulated: black); right: 2d projection of the state space with stable manifold of a saddle (black) separating the basins of attraction of a stable fixed point (green) and the 38-cycle (orange) corresponding to the spikes. Note that the true stable manifold is a 24d curved object, which for visualization purposes is represented here by a locally linear approximation in the shown 2d subspace.

**Bistable Duffing system** The Duffing (1918) system is a simple 2d nonlinear oscillator that can exhibit bistability between two spiral point attractors in certain parameter regimes (see Appx. H.3 for

more details). We trained a shPLRNN ($M = 2$, $H = 10$; Hess et al. (2023)) on this system using sparse teacher forcing (Mikhaeil et al., 2022), and used SCYFI (Eisenmann et al., 2023) to determine the fixed points. The basin boundary between the two spiral point attractors is the stable manifold of a saddle node in the center (Fig. 4A), and – as computed by our algorithm – agrees with the trajectory flows of the *true* system in blue, as further confirmed by $\Delta_\sigma \approx 0.97$ (Appx. Table 2 & Fig.6B).

**Multistable choice paradigm**   Simple models of decision making in the brain assume multistability between several choice-specific attractor states, to which the system's state is driven as one or the other choice materializes (Wang, 2002). We trained an ALRNN ($M = 15$, $P = 6$) (Brenner et al., 2024a) to perform a simple 2-choice decision making task taken from (Gerstner et al., 2014), and, as before, use SCYFI to find fixed points and Algorithm 1 to determine the stable manifold of a saddle separating the two basins of attraction corresponding to the two choices. The basin boundary consists of different planar and curved pieces and is approximately visualized in Fig. 4B in a 3d subspace (of the 15d system), together with the reconstructed system's fixed points (green) and some trajectories of the *true* system in blue ($\Delta_\sigma \approx 0.95$ in this case, Appx. Table 2 & Fig. 6C). While visualization becomes tricky in these higher-dimensional cases, computing distances to the manifold, using the construction in Appx. D, provides further information: In this example, we find that both stable fixed points are about equally far from the basin boundary ($d \approx 0.34$ vs. $d \approx 0.32$), giving the two possible choices about equal weight.

**Lorenz-63 attractor**   The Lorenz-63 model of atmospheric convection is probably the most famous example of a chaotic attractor. We reconstruct this system with a shPLRNN ($M = 3$, $H = 20$). Besides the chaotic attractor (blue) and two unstable spiral points (green), the system has a saddle node for which we compute the stable and unstable manifolds ($\Delta_\sigma \approx 0.78$, Appx. Table 2 & Fig.6D). Fig. 4C confirms that the manifolds computed by Algorithm 1 for the trained shPLRNN (left), agree even well with those determined by numerical integration of the *original* Lorenz ODE system (right). This example also proves that the shPLRNN faithfully captured the geometrical structure of the state space, beyond just reconstruction of the chaotic attractor itself.

**Empirical example: Single cell recordings**   In Fig. 4D we trained an ALRNN ($M = 25$, $P = 6$) on membrane potential recordings from a cortical neuron (Hertäg et al., 2012). The trained ALRNN contains a 38-cycle which corresponds to the rhythmic spiking activity in the real cell. In addition it has a stable fixed point and a saddle whose stable manifold (determined by our algorithm, $\Delta_\sigma \approx 0.76$) separates the stable cycle from the stable fixed point, as illustrated in Fig. 4D (right), where we visualized the manifold through a locally linear approximation. Although we can compute the full global 24-dimensional manifold, its inherent curvature in the 25 dimensional state space makes it impossible to visualize it directly. Computing distances to the manifold (Appx. D), however, we find that the oscillation at its lowest point is less sensitive to small perturbations than the cell's resting state, with a minimum distance of $d \approx 2.3$ to the basin boundary, compared to $d \approx 1.2$ for the fixed point. Many types of cortical cells exhibit this type of bistability between spiking activity and a stable equilibrium near the resting or a more depolarized potential (Izhikevich, 2007; Durstewitz and Gabriel, 2007), and this example demonstrates how our algorithm can be utilized to reveal the structure of the state space supporting this type of dynamics from real cells.

## 5   HOMO-/HETEROCLINIC ORBITS AND DETECTION OF CHAOS

The stable and unstable manifolds can be used to identify homoclinic and heteroclinic orbits, as defined in Sect. 3.2 (Def. 3, 4). Intersections between the stable and unstable manifolds of a saddle $p$ lead to homoclinic points, or to heteroclinic points of two saddles $p \neq q$. The existence of such points inevitably gives rise to a complex fractal geometry (a so-scalled horseshoe structure, see Appx. I) and thus chaos due to the Smale-Birkhoff Homoclinic Theorem (Wiggins, 1988; Simpson, 2016). Finding such intersections is therefore highly illuminating for determining the dynamical behavior of a system. This is illustrated for a simple 2d PL map in Fig. 5A, which shows the homoclinic intersections identified by Algorithm 1. For this 2d case, we can in fact analytically determine the presence of homoclinic points, as worked out in Appx. H.2, the results of which agree with Algorithm 1. Fig. 5B illustrates the whole resulting chaotic attractor, which happens to lie on the unstable manifold in this case. Fig. 5C provides the bifurcation diagram as one varies the model's bias term as a control parameter, and Fig. 5D the system's two Lyapunov exponents across the chaotic range

of $h_1$ (confirming the presence of 'robust chaos', as the Lyapunov exponents do not change across the chaotic regime (Banerjee et al., 1998)). Finally, in Appx. Fig. 10 we show the identification of a heteroclinic intersection by our algorithm in a high-dimensional empirical example, human electrocardiograms (ECG), thus confirming the presence of chaos in this signal.

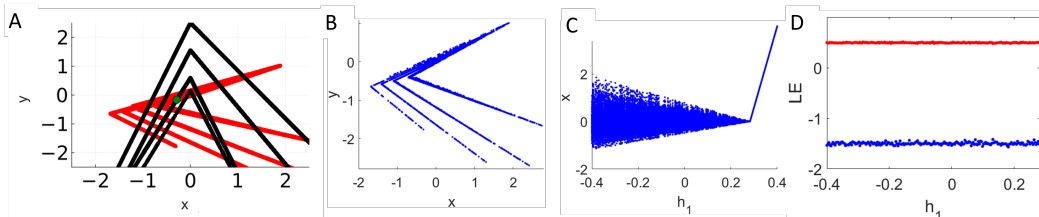

Figure 5: A) Stable (black) and unstable (red) manifolds of a saddle point (green dot) and their homoclinic intersections as identified by our algorithm. B) Structure of the chaotic attractor caused by these homoclinic intersections. C) Bifurcation diagram as a function of bias parameter $h_1$. D) Lyapunov exponents across the $h_1$-range for which the chaotic attractor exists.

## 6 CONCLUSIONS

Here we presented a novel semi-analytical algorithm for determining stable and unstable manifolds of fixed and cyclic points in ReLU-based RNNs. Within each linear subregion, the algorithm provides an analytical construction for the manifold segments, and then recursively assembles these segments into the global un-/stable manifolds by forward- or backward-iterating the RNN map $F_\theta$ across subregions. These manifolds are profoundly important for studying an RNN's dynamical repertoire, illuminating dynamical mechanisms in an underlying system reconstructed by the RNN, or the mechanisms by which an RNN solves a given task. Stable manifolds of saddle points segregate the state space into different basins of attraction, giving rise to the computationally important property of multistability (Durstewitz et al., 2000; Feudel et al., 2018). Intersections of stable and unstable manifolds, in turn, lead to homo- or heteroclinic points which produce a fractal geometry and chaos (Wiggins, 1988). While here we developed these algorithms for the class of PLRNNs (Brenner et al., 2022; Hess et al., 2023; Brenner et al., 2024a), extensions to other PL systems, like SLDS (Linderman et al., 2017; 2016) or TLS (Parmelee et al., 2022; Morrison et al., 2024; Hahnloser and Seung, 2000) are conceivable. SLDS, unlike PLRNNs, are not continuous across boundaries of linear regions, however, rendering them less suitable for DSR in general and making it more tricky to impose invertibility. TLNs, on the other hand, are defined in continuous rather than discrete time. Techniques for converting discrete into continuous time PLRNNs may thus be of help here (Monfared and Durstewitz, 2020). Sometimes restrictions are imposed on the weight matrices (like binary weights or mutual inhibition), leading to nice theoretical results (Curto and Morrison, 2023; Parmelee et al., 2022; Morrison et al., 2024) but also curtailing the model's flexibility for approximating arbitrary DS.

**Limitations** Limitations arise in the context of chaotic dynamics, where invariant manifolds may fold into fractal structures. While points from these manifolds may still be sampled, the analytic construction through curved/planar segments spanned by support vectors breaks down, as the intricate, self-similar geometry of fractals cannot be captured this way. Nevertheless, we still may be able to retrieve some important dynamical characteristics by determining homo- or heteroclinic intersections, as discussed in Section 5. Another limitation is that in the worst case scenario the algorithm may scale as $2^P$ with the number of linear subregions $P$. However, as shown in (Brenner et al., 2024a), the number of subregions utilized by trained PLRNNs quickly saturates, suggesting an at most polynomial scaling if one restricts attention to the domain explored by the data (see also Fig. 9). Finally, although not strictly part of the present algorithm, whether one can recover all relevant manifolds also depends one whether all saddle points were detected in the first place, of course. In any case, we emphasize that this is the *first* algorithm for detecting un-/stable manifolds in ReLU-based RNNs.

**Code** is available at `https://github.com/DurstewitzLab/DetectingManifolds`.

## 7 ACKNOWLEDGEMENTS

This work was primarily supported by Samsung Advanced Institute of Technology, Samsung Electronics Co., Ltd. Additional funding was provided through the German Research Foundation (DFG) within the FOR-5159 research cluster (Du 354/14-1), the collaborative research center TRR 265, subproject A06 and by the German Ministry for Education & Research (BMBF) within the FEDORA (01EQ2403F) consortium. Z.M. is also grateful to the Bundesministerium für Forschung, Technologie und Raumfahrt (BMFTR, Federal Ministry of Research, Technology and Space) for funding through project OIDLITDSM, No. 01IS24061.

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

APPENDIX:

## A   LLM USAGE AND CODE AVAILABILITY

Code for the algorithms developed here and results produced is available at `https://github.com/DurstewitzLab/DetectingManifolds`. Large Language Models (LLMs) were used for implementation of standard routines and assistance with literature search.

## B   DERIVATION OF EQ. 4

The stable and unstable manifolds of saddles are invariant sets with all trajectories with initial conditions on the manifold confined to the manifold for all $t$ (Alligood et al., 1996; Perko, 2001), $F_\theta(W^\sigma) \subseteq W^\sigma$. We recursively assemble these manifolds following Eq. 3, with points sampled from $W^\sigma$ following Eq. 4 within each linear subregion. Here we derive this equation starting with a linear system of the form $z_{t+1} = L z_t$.

**Case (1)**: If the geometric multiplicity equals the algebraic multiplicity, then $L$ is diagonalizable and admits a basis of linearly independent eigenvectors $v_1, \cdots, v_n$. Thus, the orbits of the system can be expanded in terms of eigenvectors as

$$z_t = c_1 \lambda_1^t v_1 + c_2 \lambda_2^t v_2 + \cdots + c_n \lambda_n^t v_n. \tag{7}$$

**Case (2)**: If the geometric multiplicity of $\lambda_i$ is less than the algebraic multiplicity, then the eigenvalue is said to be defective and $L$ admits a basis of generalized eigenvectors. In this case, $L$ has a Jordan block $J_d(\lambda_i)$ of size $d > 1$

$$J_d(\lambda_i) = \begin{pmatrix} \lambda_i & 1 & 0 & \dots & 0 \\ 0 & \lambda_i & 1 & \dots & 0 \\ 0 & 0 & \lambda_i & \dots & 0 \\ \vdots & \vdots & \vdots & \ddots & 1 \\ 0 & 0 & 0 & \dots & \lambda_i \end{pmatrix},$$

with the $t$-th power

$$J_m^t(\lambda_i) = \begin{pmatrix} \lambda_i^t & \binom{t}{1} \lambda_i^{t-1} & \binom{t}{2} \lambda_i^{t-2} & \cdots & \binom{t}{d-1} \lambda_i^{t-d+1} \\ 0 & \lambda_i^t & \binom{t}{1} \lambda_i^{t-1} & \cdots & \binom{t}{d-2} \lambda_i^{t-d+2} \\ \vdots & \vdots & \ddots & \ddots & \vdots \\ 0 & 0 & \dots & \lambda_i^t & \binom{t}{1} \lambda_i^{t-1} \\ 0 & 0 & \dots & 0 & \lambda_i^t \end{pmatrix}.$$

Moreover, a generalized eigenvector $w_d \neq 0$ of degree $d$, corresponding to the defective eigenvalue $\lambda_i$, satisfies

$$(L - \lambda I)^d w_d = 0, \qquad \text{but} \quad (L - \lambda I)^{d-1} w_d \neq 0, \tag{8}$$

and $L$ has $d$ linearly independent generalized eigenvectors associated with $\lambda_i$. In fact, we can construct a chain of generalized eigenvectors $\{w_1, \cdots, w_d\}$ such that

$$(L - \lambda I) w_d = w_{d-1}, \; (L - \lambda I) w_{d-1} = w_{d-2}, \; \cdots, \; (L - \lambda I) w_2 = w_1, \tag{9}$$

where $w_1$ is a regular eigenvector. Given a chain of length $d$, the orbit contribution corresponding to the Jordan block $J_d(\lambda_i)$ is given by

$$z_t^{\text{defective}} = \lambda_i^t \Big( c_1 w_1 + c_2 \, t w_2 + c_3 \frac{t^2}{2!} w_3 + \cdots + c_d \frac{t^{d-1}}{(d-1)!} w_d \Big). \tag{10}$$

The full orbit of the system is a linear combination of contributions from the non-defective eigenvalues $\lambda_j$, $1 \leq j \neq i \leq n$, and the defective eigenvalue $\lambda_i$ according to

$$z_t = \sum_{\substack{j=1 \\ j \neq i}}^{n} c_j \lambda_j^t v_j + \lambda_i^t \sum_{r=1}^{d} c_r \frac{t^{r-1}}{(r-1)!} w_r. \tag{11}$$

This formula describes exponential evolution for all non-defective eigenvalues and polynomially modified exponentials for the defective eigenvalue $\lambda_i$.

A PLRNN is not strictly linear, but affine in each subregion, see Eq. 2. For an affine system of the form $z_{t+1} = Lz_t + h$, Eq. 11 consists of a homogeneous part (determined by the eigenstructure of $L$) and a particular part due to the constant bias term $h$. In the presence of a defective eigenvalue $\lambda_i$ with Jordan block of size $d$, the full orbit is given by

$$z_t = \sum_{\substack{j=1 \\ j \neq i}}^{n} c_j \lambda_j^t v_j \;+\; \lambda_i^t \sum_{r=1}^{d} c_r \frac{t^{r-1}}{(r-1)!} w_r \;+\; \sum_{k=0}^{t-1} L^k h. \tag{12}$$

The final term $\sum_{k=0}^{t-1} L^k h$ accounts for the cumulative effect of the bias, modifying the orbit away from purely exponential or polynomial-exponential behavior.

## C  ADDITIONAL DETAILS ON MANIFOLD CONSTRUCTION

**Backtracking algorithm and other PLRNN variants**    Step 2 in sect. 3.3 outlines the steps for inverting the RNN map $F_\theta$ for PLRNNs of the form Eq. 1, spelled out in detail in Algorithm 2. This can easily be amended for other ReLU-based RNNs. For instance, for a 1-hidden layer PLRNN, called the shallow PLRNN (shPLRNN) (Hess et al., 2023), given by

$$z_{t+1} = Az_t + W_1 D_t(W_2 z_t + h_2) + h_1 \tag{13}$$

where $A \in \mathbb{R}^{M \times M}$, $W_1 \in \mathbb{R}^{M \times H}$, $W_2 \in \mathbb{R}^{H \times M}$, $h_2 \in \mathbb{R}^H$, $h_1 \in \mathbb{R}^M$, the inversion of this map yields

$$z_{t-1} = (A + W_1 D_{t-1} W_2)^{-1}(z_t - W_1 D_{t-1} h_2 - h_1) \tag{14}$$

**Manifold modeling**    To generate initial conditions on the local manifold in Algorithm 1, we sample perturbations in the stable eigenspace around a base point $x_b$ (equal to $p$ in the initial linear subregion). To account for anisotropy in the vector field and potential undersampling of slow eigendirections, we first construct an orthonormal basis $E_{\text{ortho}}^\sigma$ via the QR decomposition ($E^\sigma = E_{\text{ortho}}^\sigma R$) which preserves the span of the stable eigenspace while avoiding numerical degeneracy. Transforming the linear operator into this basis yields the *effective* eigenvalues $\tilde{\lambda}_i = \left(R \operatorname{diag}(\lambda_1, \ldots, \lambda_{d_s}) R^{-1}\right)_{ii}$, which govern contraction rates along the orthonormalized directions. Because systems with disparate timescales typically exhibit large variability along fast eigendirections but very limited variability along slow ones, naive sampling leads to poorly conditioned point clouds. We therefore rescale perturbations inversely to the magnitude of the effective eigenvalues via $s_i = \frac{1}{|\tilde{\lambda}_i|^c}$, where the hyperparameter $c$ controls how strongly slow directions are emphasized. We then draw $N_s$ perturbation coefficients from a Gaussian mixture model (GMM), $\alpha \sim \frac{1}{3} \sum_{k=1}^{3} \mathcal{N}\left(0, \sigma_k^2 I\right)$, $\sigma_k \in \{0.01, 0.1, 0.5\}$, where the three scales $\sigma_k$ ensure multi-scale coverage of the local manifold. Eigenvalue-based rescaling is applied multiplicatively ($\tilde{\alpha}_i = s_i \alpha_i$). Finally, the sampled points on the manifold are generated by $x = x_b + E_{\text{ortho}}^\sigma \tilde{\alpha}$, and points violating ReLU-constraints (i.e., are inconsistent with the active set at $x_b$) are rejected. This procedure yields a numerically well-conditioned set of initial points distributed across all stable eigendirections and across multiple sampling scales.

In the case of curved manifolds, we used kernel-PCA (Schölkopf et al., 1998) for approximation in Algorithm 1. Specifically, we used an RBF kernel, the squared exponential kernel $K(x, y) = \exp(-\frac{1}{2}\|x - y\|^2)$, for its generally good approximation properties for smooth manifolds with curvature as described by 4 (Hofmann et al., 2007). Since $K(x, y)$ decays with Euclidean distance, kernel-PCA with an RBF kernel emphasizes local geometry, which is suitable for manifolds with curvature. Moreover, the RBF kernel is universal and capable of representing smooth curvature. While more specialized kernels (e.g., spectral–periodic or polynomial–exponential) may sound like a theoretically better match, they tend to require careful tuning of system-specific frequencies or decay rates. The RBF kernel, by contrast, offers a robust and broadly applicable alternative that performs reliably without region-dependent parameter adjustments.

---

**Algorithm 2** Backtracking Time Series in a ReLU based RNNs

---

1: $z_T \leftarrow$ an initial State
2: $\theta \leftarrow$ Parameters
3: Initialize list: $S = [z_T]$
4: **for** $t = T : 1$ **do**
5:     $z_t = S[T - t]$
6:     $D_{t-1} \leftarrow \text{diag}(z_t > 0)$                             $\triangleright$ Initialize D as a diagonal matrix
7:     $z_{t-1}^* \leftarrow F^{-1}(\theta, D_{t-1}, z_t)$                        $\triangleright$ Perform a backward step
8:     $\tilde{z}_t \leftarrow F(\theta, z_{t-1}^*)$                              $\triangleright$Perform a forward step
9:     **if** $\tilde{z}_t = z_t$ **then**
10:         $S \leftarrow S \cup \{z_{t-1}^*\}$                      $\triangleright$If forward step is correct
11:     **else**
12:         $D_{t-1} \leftarrow \text{diag}(z_{t-1}^* > 0)$               $\triangleright$Update D with new candidate
13:         $z_{t-1}^* \leftarrow F^{-1}(\theta, D_{t-1}, z_t)$              $\triangleright$ Retry backward step
14:         $\tilde{z}_t \leftarrow F(\theta, z_{t-1}^*)$                     $\triangleright$ Retry forward step
15:         **if** $\tilde{z}_t = z_t$ **then**
16:             $S \leftarrow S \cup \{z_{t-1}^*\}$                 $\triangleright$ If forward step is correct
17:         **else**
18:             $z_{t-1}^* \leftarrow$TryPreviousRegions$(\theta, D_{\Omega_{pool}}, z_t)$     $\triangleright$ Try prev. regions $D_{\Omega_{pool}}$
19:            **if** $z_{t-1}^* = $ none **then**
20:                $z_{t-1}^* \leftarrow$TryBitflips$(\theta, z_t)$            $\triangleright$ Iteratively check neighbours
21:                **if** $z_{t-1}^* \neq$ none **then**
22:                    $S \leftarrow S \cup \{z_{t-1}^*\}$
23:                **else**
24:                    **return** $S$
25:                **end if**
26:            **else**
27:                $S \leftarrow S \cup \{z_{t-1}^*\}$
28:            **end if**
29:         **end if**
30:     **end if**
31: **end for**
32: **return** $S$

---

---

1: **function** BACKWARDFORWARD$(\theta, D, z)$
2:     $z^* \leftarrow F^{-1}(\theta, D, z)$
3:     $\tilde{z} \leftarrow F(\theta, z^*)$
4:     **return** $z^*, \tilde{z}$
5: **end function**
6: **function** TRYPREVIOUSREGIONS$(\theta, \boldsymbol{D}\_pool, \boldsymbol{z})$
7:     **for** $\boldsymbol{D} \in \boldsymbol{D}\_$pool **do**
8:         $\boldsymbol{z}^*, \tilde{\boldsymbol{z}} \leftarrow$ BackwardForward$(\theta, \boldsymbol{D}, \boldsymbol{z})$
9:         **if** $\tilde{\boldsymbol{z}} = \boldsymbol{z}$ **then**
10:             **return** $\boldsymbol{z}^*$
11:         **end if**
12:     **end for**
13:     **return** none
14: **end function**
15: **function** TRYBITFLIPS$(\theta, \boldsymbol{z})$
16:     $\boldsymbol{D} = \text{diag}(\boldsymbol{z} > 0)$
17:     **for** $k = 1 : \text{num\_relus}$ **do**
18:         $\boldsymbol{D}\_$versions $\leftarrow$ generate_bitflip$(k, \boldsymbol{D})$         $\triangleright$ all possible bitflips of order $k$
19:         **for** $\boldsymbol{D}_i \in \boldsymbol{D}\_$versions **do**
20:             $\boldsymbol{z}^*, \tilde{\boldsymbol{z}} \leftarrow$ BackwardForward$(\theta, \boldsymbol{D}_i, \boldsymbol{z})$
21:             **if** $\tilde{\boldsymbol{z}} = \boldsymbol{z}$ **then**
22:                **return** $\boldsymbol{z}^*$
23:             **end if**
24:         **end for**
25:     **end for**
26:     **return** none

18

27: **end function**

---

**Fallback algorithm**   We are usually interested in the structure of the state space in the context of DS reconstruction where we train RNNs on time series observations from time-continuous DS. In these cases, Algorithm 1 is always applicable. More generally, however, maps may exhibit large jumps, with orbits "erratically" hopping around among subregions. In these cases the un-/stable manifolds may acquire a more complicated, non-smooth geometry (as in Fig. 3B & Fig. 5). Algorithm 3 formulates a 'fallback' procedure for such cases that works by perturbing seed points along the analytically defined local manifold, and then iterates $F_\theta$ to generate a larger set of support vectors. As manifolds can re-enter the same subregion in multiple folds, we apply HDBSCAN (McInnes et al., 2017; Ester et al., 1996) to cluster support vectors into distinct segments. Although computationally more demanding, this fallback reliably captures manifolds with discontinuous or folding structures when sequential tracing as in Algorithm 1 fails.

---

**Algorithm 3** Finding stable/unstable manifolds: fallback algorithm

---
1: $(\boldsymbol{p}, E) \leftarrow$ SCYFI $\qquad\qquad\qquad\qquad\qquad\qquad \triangleright \boldsymbol{p}$: Fixed Point, E: Eigenvectors
2: $S^{\{+1\}} \leftarrow \emptyset \qquad\qquad\qquad\qquad\qquad\qquad\qquad\qquad\qquad\qquad \triangleright$ Stable Manifolds
3: $S^{\{-1\}} \leftarrow \emptyset \qquad\qquad\qquad\qquad\qquad\qquad\qquad\qquad\qquad\qquad \triangleright$ Unstable Manifolds
4: **for** i=1:$N_1$ **do**
5: $\qquad z_0 = \boldsymbol{p} \qquad\qquad\qquad\qquad\qquad\qquad\qquad\qquad \triangleright$ For $N_1$ different initialisations
6: $\qquad$ **for** $v^{\{-1\}} \in E^{\{-1\}}$ **do**
7: $\qquad\qquad z_0 \mathrel{+}= v^{\{-1\}} \cdot rand() \qquad\qquad\qquad\qquad\qquad\qquad \triangleright$ Perturb into subspace
8: $\qquad$ **end for**
9: $\qquad T^{\{-1\}} \leftarrow GetForwardTS(z_0)$
10: $\qquad S^{\{-1\}} \leftarrow S^{\{-1\}} \cup \{T^{\{-1\}}\}$
11: **end for**
12: **for** i=1:$N_2$ **do**
13: $\qquad z_0 = \boldsymbol{p} \qquad\qquad\qquad\qquad\qquad\qquad\qquad\qquad \triangleright$ For $N_2$ different initialisations
14: $\qquad$ **for** $v^{\{+1\}} \in E^{\{+1\}}$ **do**
15: $\qquad\qquad z_0 \mathrel{+}= v^{\{+1\}} \cdot rand() \qquad\qquad\qquad\qquad\qquad\qquad \triangleright$ Perturb into subspace
16: $\qquad$ **end for**
17: $\qquad T^{\{+1\}} \leftarrow GetBackwardTS(z_0)$
18: $\qquad S^{\{+1\}} \leftarrow S^{\{+1\}} \cup \{T^{\{+1\}}\}$
19: **end for**
20: $\widetilde{S}^{\{+1\}} \leftarrow \emptyset \qquad\qquad\qquad\qquad\qquad\qquad\qquad \triangleright$ Piecewise linear manifold fits
21: $\widetilde{S}^{\{-1\}} \leftarrow \emptyset$
22: **for each** $D \in D_\Omega$ **do**
23: $\qquad S_\Omega^{\{+1\}} \leftarrow S^{\{+1\}} \cap D_\Omega \qquad\qquad\qquad\qquad\qquad\qquad \triangleright$ Go through all subregions
24: $\qquad S_\Omega^{\{-1\}} \leftarrow S^{\{-1\}} \cap D_\Omega$
25: $\qquad (C_\Omega^{\{+1\}}, C_\Omega^{\{-1\}}) \leftarrow \text{FIT}((S_\Omega^{\{+1\}}, S_\Omega^{\{-1\}})) \qquad\qquad\qquad \triangleright$ Cluster points and fit
26: $\qquad \widetilde{S}^{\{+1\}} \leftarrow \widetilde{S}^{\{+1\}} \cup \{C_\Omega^{\{+1\}}\}$
27: $\qquad \widetilde{S}^{\{-1\}} \leftarrow \widetilde{S}^{\{-1\}} \cup \{C_\Omega^{\{-1\}}\}$
28: **end for**
29: **return** $(\widetilde{S}^{\{+1\}}, \widetilde{S}^{\{-1\}}) \qquad\qquad\qquad\qquad\qquad \triangleright$ Piecewise linear manifolds

---

# D   ALTERNATIVE CONSTRUCTION OF MANIFOLDS

Algorithm 1 constructs the local manifolds within each subregion using PCA or kernel-PCA. Here we provide an alternative parameterization that may be simpler to use for some purposes, like calculating distances on and to the manifold. First note that Eq. 4 (cf. Appx. B) yields a 1d curve per initial condition. An invariant $d$-dimensional manifold ($d > 1$) is a smooth surface that contains infinitely many such curves. To describe the entire manifold, we need a parameterization using $d$ variables, rather than a family of 1-dimensional curves.

Let the state space be partitioned into finitely many subregions $\{\mathcal{S}_k\}_{k=1}^K$ and

$$F(\boldsymbol{x}) = \boldsymbol{L}_k \boldsymbol{x} + \boldsymbol{m}_k, \qquad \boldsymbol{x} \in \mathcal{S}_k. \qquad (15)$$

Assume that there exists a $d$-dimensional invariant manifold $W^{\pm 1}$ in $1 < \mathbb{K}_I \subseteq \{1, 2, \ldots, K\}$ subregions of $F$, and for every such subregion

$$W_k^{\pm 1} := W^{\pm 1} \cap \mathcal{S}_k \neq \emptyset, \qquad k \in \mathbb{K}_I. \tag{16}$$

We focus on a single fixed point $\boldsymbol{p}$ of $F$, and assume that this fixed point lies in a unique subregion $\mathcal{S}_{k_0}$. The invariant manifold $W^{\pm 1}$ is the un-/stable manifold associated with this fixed point. Accordingly, the local manifold segment used to initiate the construction is centered at this fixed point. The manifold segments $W_k^{\pm 1}$, $k \in \mathbb{K}_I \setminus \{k_0\}$, in other subregions are obtained by propagating this local manifold under the dynamics of $F$. Thus, in all other subregions the local manifold segments are generally not centered at a fixed point of the corresponding affine subsystem.

For each subregion $\mathcal{S}_k$, choose a reference point

$$\boldsymbol{x}_k^\star \in W_k^{\pm 1}, \tag{17}$$

which is assumed to lie on the invariant manifold and serves as a local anchor of the manifold segment in subregion $k$. For instance, $\boldsymbol{x}_k^\star$ can be chosen as a point where a single invariant 1d trajectory generated by Eq. 4 intersects the subregion $\mathcal{S}_k$. In the unique subregion that contains the fixed point, $\boldsymbol{x}_k^\star$ coincides with this fixed point.

Suppose that $T_k \in GL(n, \mathbb{R})$ is any invertible linear change of coordinates whose first $d$ columns span the tangent space $T_{x_k^\star} W^{\pm 1}$ of the invariant manifold at $\boldsymbol{x}_k^\star$. Denote by $\mathcal{S}_{k_0}$ the unique subregion that contains the fixed point $\boldsymbol{p}$ and set $\boldsymbol{x}_{k_0}^\star = \boldsymbol{p}$. At $\boldsymbol{p}$, the tangent space of the $d$-dimensional invariant manifold $W^{\pm 1}$ is given by the corresponding invariant eigenspace of $\boldsymbol{L}_{k_0}$ (stable or unstable). Let

$$\boldsymbol{V}_{k_0} = \left[ \boldsymbol{v}_{k_0}^{(1)}, \ldots, \boldsymbol{v}_{k_0}^{(d)} \right] \in \mathbb{R}^{n \times d} \tag{18}$$

be a basis matrix of this eigenspace. For any subregion $\mathcal{S}_k$, let $\boldsymbol{x}_k^\star \in W_k^{\pm 1}$ be an anchor point on the invariant manifold. Assume that there exists a finite sequence of points

$$\boldsymbol{x}_0 = \boldsymbol{p}, \ \boldsymbol{x}_1, \boldsymbol{x}_2, \cdots, \ \boldsymbol{x}_{N_k} = \boldsymbol{x}_k^\star \tag{19}$$

such that

$$\boldsymbol{x}_{t+1} = F(\boldsymbol{x}_t), \qquad t = 0, \cdots, N_{k-1}, \tag{20}$$

and $\boldsymbol{x}_t \in W^{\pm 1}$ for all $t$. In particular, such a sequence can be obtained by iterating a point on an invariant 1-dimensional trajectory generated by Eq. 4, but the construction does not depend on how the points on $W^{\pm 1}$ are generated, provided that they follow the dynamics. Then the tangent space of the invariant manifold at $\boldsymbol{x}_k^\star$ is obtained by transporting the tangent space at $\boldsymbol{p}$ along this trajectory

$$\boldsymbol{V}_k = \boldsymbol{L}_{r(N_k - 1)} \cdots \boldsymbol{L}_{r(1)} L_{r(0)} \, \boldsymbol{V}_{k_0}. \tag{21}$$

The columns of $\boldsymbol{V}_k$ span $T_{\boldsymbol{x}_k^\star} W^{\pm 1}$. The matrix $\boldsymbol{T}_k$ is obtained by taking the first $d$ columns equal to the columns of $\boldsymbol{V}_k$ and completing them with any additional $(n - d)$ linearly independent vectors to form an invertible matrix.

A $d$-dimensional un-/stable manifold segment $W_k^{\pm 1}$ is itself a $d$-dimensional smooth surface embedded in $\mathbb{R}^n$. The *Implicit Function Theorem* guarantees that any $d$-dimensional smooth manifold that is transversal to a complementary $(n - d)$-dimensional subspace, can be locally expressed as a graph of a smooth function $h_k : \mathbb{R}^d \to \mathbb{R}^{n-d}$.

We introduce local coordinates $(\boldsymbol{u}, \boldsymbol{v})$ by

$$\boldsymbol{x} = \boldsymbol{x}_k^\star + \boldsymbol{T}_k \begin{pmatrix} \boldsymbol{u} \\ \boldsymbol{v} \end{pmatrix}, \qquad \boldsymbol{u} \in \mathbb{R}^d, \ \boldsymbol{v} \in \mathbb{R}^{n-d}. \tag{22}$$

In these coordinates, the manifold segment in subregion $k$ is represented by a graph

$$W_k^{\pm 1} = \left\{ \boldsymbol{x}_k^\star + \boldsymbol{T}_k \begin{pmatrix} \boldsymbol{u} \\ h_k(\boldsymbol{u}) \end{pmatrix} : \boldsymbol{u} \in U_k \right\}, \tag{23}$$

with

$$h_k(0) = 0, \qquad Dh_k(0) = 0. \tag{24}$$

The condition $Dh_k(0) = 0$ indicates that the chosen coordinates are tangential to the invariant manifold at the anchor point $x_k^\star$.

For $x \in \mathcal{S}_k$ we have

$$x^+ = F(x) = L_k x + m_k. \tag{25}$$

Using the coordinate change associated with subregion $k$,

$$x = x_k^\star + T_k \begin{pmatrix} u \\ v \end{pmatrix}, \tag{26}$$

the image of $(u, v)$ under the dynamics can be written as

$$\begin{pmatrix} u^+ \\ v^+ \end{pmatrix} = T_k^{-1} L_k T_k \begin{pmatrix} u \\ v \end{pmatrix} + T_k^{-1}(L_k x_k^\star + m_k - x_k^\star). \tag{27}$$

We denote the block decomposition by

$$T_k^{-1} L_k T_k = \begin{pmatrix} A_k & B_k \\ C_k & D_k \end{pmatrix}, \qquad T_k^{-1}(L_k x_k^\star + m_k - x_k^\star) = \begin{pmatrix} a_k \\ b_k \end{pmatrix}. \tag{28}$$

If both $x$ and $F(x)$ belong to the same subregion $\mathcal{S}_k$, then points on the manifold satisfy

$$v = h_k(u), \qquad v^+ = h_k(u^+). \tag{29}$$

Substituting $v = h_k(u)$ into Eq. 27 yields

$$u^+ = A_k u + B_k h_k(u) + a_k, \tag{30}$$

$$v^+ = C_k u + D_k h_k(u) + b_k. \tag{31}$$

Hence, the local invariance equation in subregion $k$ is given by

$$h_k(A_k u + B_k h_k(u) + a_k) = C_k u + D_k h_k(u) + b_k. \tag{32}$$

The affine terms $a_k$ and $b_k$ appear only for $x_k^\star \neq p$, that is, when the anchor $x_k^\star$ is not a fixed point of the local affine map.

Let $x \in W_k^{\pm 1} \subset \mathcal{S}_k$ and

$$F(x) \in \mathcal{S}_j, \qquad j \neq k. \tag{33}$$

In subregion $j$ the manifold is represented by

$$x = x_j^\star + T_j \begin{pmatrix} \tilde{u} \\ h_j(\tilde{u}) \end{pmatrix}. \tag{34}$$

Invariance of the global manifold implies the existence of a local reparameterization map

$$\tilde{u} = R_{k \to j}(u) \tag{35}$$

such that

$$x_j^\star + T_j \begin{pmatrix} R_{k \to j}(u) \\ h_j(R_{k \to j}(u)) \end{pmatrix} = L_k \left( x_k^\star + T_k \begin{pmatrix} u \\ h_k(u) \end{pmatrix} \right) + h_k. \tag{36}$$

Eq. 36 couples the local graph parameterizations of neighboring subregions.

Differentiating Eq. 36 at $u = 0$ and using $Dh_k(0) = 0$, $Dh_j(0) = 0$, yields

$$T_j \begin{pmatrix} DR_{k \to j}(0) \\ 0 \end{pmatrix} = L_k T_k \begin{pmatrix} I_d \\ 0 \end{pmatrix}. \tag{37}$$

Since $T_k$ and $T_j$ are invertible, $DR_{k \to j}(0)$ has full rank $d$. Therefore, the intrinsic dimension of the invariant manifold is preserved when passing from subregion $k$ to subregion $j$, in agreement with the assumption that the manifold cannot change its dimensionality across subregions.

**Global manifold.** The invariant manifold of the piecewise-linear system is obtained as the union of the locally parameterized segments

$$W^{\pm 1} = \bigcup_{k=1}^{K} \left\{ x_k^\star + T_k \begin{pmatrix} u \\ h_k(u) \end{pmatrix} : u \in U_k \right\}, \tag{38}$$

with consistency between neighbouring subregions enforced by Eq. 36.

**How to compute $h_k$ in practice**  We outline the standard series method which produces a polynomial approximation of $h_k(\boldsymbol{u})$ for each subregion $\mathcal{S}_k$. This method works for either stable or unstable manifolds by choosing the corresponding tangent space basis $\boldsymbol{T}_k$ and block matrices $(\boldsymbol{A}_{kj}, \boldsymbol{B}_{kj}, \boldsymbol{C}_{kj}, \boldsymbol{D}_{kj})$, which link region $k$ to its neighboring region $j = r(k)$ following the manifold flow. The affine terms $\boldsymbol{a}_k, \boldsymbol{b}_k$, appear when the anchor point $\boldsymbol{x}_k^*$ is not a fixed point of the local affine map.

**Step 1:** In each subregion $\mathcal{S}_k$, develop $h_k(\boldsymbol{u})$ as a multivariate Taylor series about $\boldsymbol{u} = 0$ (no linear term because of tangency to the first $d$ columns of $\boldsymbol{T}_k$ at the anchor point $\boldsymbol{x}_k^*$):

$$h_k(\boldsymbol{u}) = \sum_{|\alpha| \geq 2} h_{k,\alpha}\, \boldsymbol{u}^\alpha, \tag{39}$$

where $\alpha \in \mathbb{N}^m$ is a multi-index and $\boldsymbol{u}^\alpha = \prod_{i=1}^m \boldsymbol{u}_i^{\alpha_i}$.

**Step 2:** Substitute the series into the invariance equation linking region $k$ to its image region $j$:

$$h_j(\boldsymbol{A}_{kj}\boldsymbol{u} + \boldsymbol{B}_{kj}h_k(\boldsymbol{u}) + \boldsymbol{a}_k) = \boldsymbol{C}_{kj}\boldsymbol{u} + \boldsymbol{D}_{kj}h_k(\boldsymbol{u}) + \boldsymbol{b}_k. \tag{40}$$

Expand both sides into multivariate Taylor series in $\boldsymbol{u}$. For the left-hand side, first expand the argument $U = \boldsymbol{A}_{kj}\boldsymbol{u} + \boldsymbol{B}_{kj}h_k(\boldsymbol{u}) + \boldsymbol{a}_k$, then expand $h_j(U)$ using its own series expansion about $U = 0$. Equate coefficients of monomials $\boldsymbol{u}^\alpha$ of homogeneous degree $|\alpha| = \ell$ to obtain equations for the unknown tensors $h_{k,\alpha}$ and $h_{j,\alpha}$.

**Step 3:** Solve order by order:

- **For order 1:** The linear terms vanish automatically due to $Dh_k(0) = 0$, $Dh_j(0) = 0$, and condition $\boldsymbol{C}_{kj} = 0$ which must hold for the manifold to exist as a graph. This serves as a consistency check.

- **For order 2:** For $|\alpha| = 2$, the left-hand side contributes $h_j^{(2)}(\boldsymbol{A}_{kj}\boldsymbol{u})^{\otimes 2}$ from the quadratic term of $h_j$, plus terms involving $\boldsymbol{a}_k$ and $\boldsymbol{B}_{kj}h_k(\boldsymbol{u})$. The right-hand side contributes $\boldsymbol{D}_{kj}h_k(\boldsymbol{u})$. This yields a linear relation:

$$\frac{1}{2}\mathsf{H}_j^{(2)}(\boldsymbol{A}_{kj}\boldsymbol{u})^{\otimes 2} = \boldsymbol{D}_{kj}\left(\frac{1}{2}\mathsf{H}_k^{(2)}\boldsymbol{u}^{\otimes 2}\right) + \text{(terms from } \boldsymbol{a}_k) \tag{41}$$

  where $\mathsf{H}_k^{(2)}$ and $\mathsf{H}_j^{(2)}$ are the quadratic coefficient tensors. Here, $(\boldsymbol{A}_{kj}\boldsymbol{u})^{\otimes 2}$ denotes the second tensor (Kronecker) power of the vector $\boldsymbol{A}_{kj}\boldsymbol{u}$, and $\mathsf{H}_j^{(2)}(\boldsymbol{A}_{kj}\boldsymbol{u})^{\otimes 2} = \mathsf{H}_j^{(2)}(\boldsymbol{A}_{kj}\boldsymbol{u},\ \boldsymbol{A}_{kj}\boldsymbol{u})$ is the evaluation of the quadratic coefficient tensor $\mathsf{H}_j^{(2)}$ (i.e., the Hessian of $h_j$ at $\boldsymbol{u} = 0$) on the pair of vectors $(\boldsymbol{A}_{kj}\boldsymbol{u},\ \boldsymbol{A}_{kj}\boldsymbol{u})$; in essence, this term 'measures' how the curvature of the manifold in region $j$ acts on the direction obtained by mapping $\boldsymbol{u}$ from region $k$ into region $j$ through the linear map $\boldsymbol{A}_{kj}$.

- **For orders $\ell \geq 3$:** The equation for $h_{k,\alpha}$ involves:
    - Known lower-order coefficients of $h_k$ (from $\boldsymbol{B}_{kj}h_k(\boldsymbol{u})$ expanding the argument)
    - Known coefficients of $h_j$ up to order $\ell - 1$ (from expanding $h_j$ about $U = 0$)
    - The affine offset $\boldsymbol{a}_k$, which shifts the expansion point and creates additional lower-order contributions

  The unknown $h_{k,\alpha}$ appears linearly, multiplied by $\boldsymbol{D}_{kj}$, yielding a linear system:

$$\mathcal{L}_{kj}(h_{k,\alpha}) = \text{RHS}_\alpha(\text{known lower-order terms}), \tag{42}$$

  where $\mathcal{L}_{kj}$ is a linear operator determined by $\boldsymbol{D}_{kj}$ and the eigenvalues of $\boldsymbol{A}_{kj}$.

**Step 4:** For each degree $\ell \geq 2$, solve the linear homological system for the coefficients $h_{k,\alpha}$. Non-resonance conditions between the eigenvalues of $\boldsymbol{A}_{kj}$ and $\boldsymbol{D}_{kj}$ ensure invertibility of the linear operator at each step. Resonances require special treatment (free parameters or solvability conditions), indicating that the manifold may not be unique or may require higher-order analysis.

**Step 5:** Repeat the process for all regions $k$ visited by the manifold, ensuring consistency at region boundaries. The final result is a piecewise polynomial representation

$$W_k^{\pm 1} = \left\{ \boldsymbol{x}_k^* + \boldsymbol{T}_k \begin{pmatrix} \boldsymbol{u} \\ h_k(\boldsymbol{u}) \end{pmatrix} : \boldsymbol{u} \in U_k \subset \mathbb{R}^d \right\} \tag{43}$$

for each subregion $\mathcal{S}_k$, with the $h_k$ given by the computed Taylor coefficients.

## E   DETAILS ON RNN TRAINING

For DS reconstruction on the simulated and empirical benchmarks (Figs. 2 & 4), we used the shPLRNN (Hess et al., 2023) and ALRNN (Brenner et al., 2024a) trained by sparse teacher forcing (Mikhaeil et al., 2022), a control-theoretically motivated training method for ensuring that reconstructed systems exhibit the same long-term statistics (cf. Appx. F) as the underlying, data-producing DS. We use the standard MSE loss

$$\ell_{\mathrm{MSE}}(\hat{\boldsymbol{X}}, \boldsymbol{X}) = \frac{1}{M \cdot T} \sum_{t=1}^{T} \|\hat{\boldsymbol{x}}_t - \boldsymbol{x}_t\|_2^2, \tag{44}$$

to which the regularization Eq. 5 was added, where $\boldsymbol{X} = \{\boldsymbol{x}_{1:T}\}$ is the training sequence of length $T$ and $\hat{\boldsymbol{X}}$ are the corresponding model predictions, obtained from the latent states through some observation function, $\hat{\boldsymbol{x}}_t = g(\boldsymbol{z}_t)$, simply taken to be the identity in the settings considered here. Sparse teacher forcing replaces latent states $\boldsymbol{z}_t$ by data-inferred states $\hat{\boldsymbol{z}}_t = g^{-1}(\boldsymbol{x}_t)$ every $\tau$ time steps via decoder model inversion. Training and experiments were performed on a single CPU (Intel Xeon Gold 6132 with 512GB RAM or Intel Xeon Gold 6248 with 832GB RAM), with parameter configurations specified in Table 1. The parameter settings are based on (Brenner et al., 2024a), refined by grid search.

| Parameter | Lorenz63 Fig. 2A | Oscillator Fig. 2B | Lorenz63 Fig. 2C | Duffing Fig. 4A | Decision making Fig. 4B | Lorenz63 Fig. 4C | Empirical Fig. 4D |
|---|---|---|---|---|---|---|---|
| Model | ALRNN | ALRNN | ALRNN | shallowPLRNN | ALRNN | shPLRNN | ALRNN |
| M | 10/20/30/50 | 40 | 30 | 2 | 15 | 3 | 25 |
| Hidden dim | - | - | - | 10 | - | 20 | - |
| #ReLUs | M - 3 | 15 | 8 | - | 6 | - | 6 |
| Sequence length | 200 | 25 | 100 | 100 | 100 | 100 | 200 |
| Gaussian noise | 0.0 | 0.0 | 0.0 | 0.0 | 0.01 | 0.05 | 0.02 |
| $\lambda_{\mathrm{invert}}$ | 0.0/0.1·exp(M) | 0.0/1e15 | 0.0/1e10 | 0.0 | 0.2 | 0.0 | 0.3 |
| Batch Size | 16 | 16 | 16 | 32 | 16 | 16 | 16 |
| Epochs | 10000 | 1000 | 1000 | 10000 | 20000 | 1000 | 2000 |
| Learning rate | 0.001 | 0.001 | 0.005 | 0.001 | 0.005 | 0.005 | 0.004 |
| $\tau$ | 16 | 10 | 15 | 15 | 15 | 15 | 20 |

Table 1: Parameter configurations. $\tau$ = teacher forcing interval.

## F   DS RECONSTRUCTION MEASURES

**Measure for Geometrical Agreement ($D_{\mathbf{stsp}}$)**   Given probability distributions $p(\boldsymbol{x})$ across ground truth trajectories and $q(\boldsymbol{x})$ across model-generated trajectories, $D_{\mathrm{stsp}}$ is defined as the Kullback-Leibler (KL) divergence

$$D_{\mathrm{stsp}} := D_{\mathrm{KL}}(p(\boldsymbol{x}) \,\|\, q(\boldsymbol{x})) = \int_{\mathbf{x} \in \mathbb{R}^N} p(\boldsymbol{x}) \log \frac{p(\boldsymbol{x})}{q(\boldsymbol{x})} \, \mathrm{d}\boldsymbol{x} \tag{45}$$

For low-dimensional observation spaces, $p(\boldsymbol{x})$ and $q(\boldsymbol{x})$ can be estimated using binning (Koppe et al., 2019; Brenner et al., 2022), yielding the discrete approximation

$$D_{\mathrm{stsp}} = D_{\mathrm{KL}}(\hat{p}(\boldsymbol{x}) \,\|\, \hat{q}(\boldsymbol{x})) \approx \sum_{k=1}^{K} \hat{p}_k(\boldsymbol{x}) \log \frac{\hat{p}_k(\boldsymbol{x})}{\hat{q}_k(\boldsymbol{x})}. \tag{46}$$

$K = m^N$ is the total number of bins, with $m$ bins per dimension. $\hat{p}_k(\boldsymbol{x})$ and $\hat{q}_k(\boldsymbol{x})$ are the normalized counts in bin $k$ for ground truth and model-generated orbits, respectively. Here we used $m = 30$ bins per dimension

**Prediction error (PE)**   The $n$-step prediction error is defined as the mean squared error between ground truth data $\{\boldsymbol{x}_t\}$ and $n$-step ahead predictions of the model $\{\hat{\boldsymbol{x}}_t\}$, i.e.

$$\text{PE}(n) = \frac{1}{N(T-n)} \sum_{t=1}^{T-n} \|\boldsymbol{x}_{t+n} - \hat{\boldsymbol{x}}_{t+n}\|_2^2. \tag{47}$$

## G   MANIFOLD QUALITY ASSESSMENT

Fig.6 illustrates the sampling process used for assessing the manifold measure (left column), Eq. 6, and the distributions of this measure (right column) when we sample points $\boldsymbol{x}_0$ from the manifold of a saddle $\boldsymbol{p}$, $\boldsymbol{x}_0 \in W^\sigma(\boldsymbol{p}) \cap \mathcal{U}$, vs. generally from some neighborhood $\boldsymbol{x}_0 \in \mathcal{U}$ as illustrated on the left. Table 2 furthermore reports for each of the examples used in the main paper the median $\tilde{\delta}_\sigma$ across values on and off the manifold (usually in tight agreement with the respective means), as well the average index $\langle I_\mathcal{U} \rangle$ for points sampled from $\mathcal{U}$, and the final quality statistic $\Delta_\sigma$ as defined in the main text.

We note that sometimes numerical issues can occur with applying Eq. 6, as the recursive iteration of $F_{\boldsymbol{\theta}}^k$ for large $k$ will exponentially diverge along unstable directions (which we always have for a saddle in both forward and backward time!) if $\boldsymbol{x}_{t+k}$ gets off track even so slightly. Eventually this will almost always be the case for $k \to \infty$, and hence the issue becomes particularly severe for stable eigendirections with absolute eigenvalues close to 1, since the slow movement in this case requires many iterations (during which errors accrue) to reach the saddle (as was the case for the empirical example in Fig. 4D). Another potential issue is that even points on the unstable manifold may temporarily move close to the saddle under forward iteration of the map; since we are looking for the least distance to the fixed point $\boldsymbol{p}$, this could further blur the differences between $\delta_\sigma$ measures of points starting on vs. off the manifold. A partial remedy might be to replace $\delta_\sigma^{\max}$ below Eq. 6 by a 'softer' threshold, like the $99^{\text{th}}$ percentile of the $\delta_\sigma$ distribution, which is a bit more lenient by allowing for some distributional outliers in the computation of $\Delta_\sigma$.

Table 2: Manifold Reconstruction Quality Statistics

| Example | On Manifold Median $\tilde{\delta}_\sigma$ | Randomly Median $\tilde{\delta}_\sigma$ | $\langle I_\mathcal{U} \rangle$ | $\Delta_{+1}$ |
|---|---|---|---|---|
| Figure 3A (left): unstable: | $1.6e-6 \pm 1.6e-6$ | $0.97 \pm 0.026$ | 0.98 | 0.98 |
| Figure 4A: Duffing | $0.0066 \pm 0.002$ | $0.46 \pm 0.25$ | 0.97 | 0.97 |
| Figure 4B: Decision making task | $0.00011 \pm 4.3e-52$ | $0.36 \pm 0.18$ | 0.95 | 0.95 |
| Figure 4C: Lorenz63 | $0.12 \pm 0.08$ | $0.8 \pm 0.2$ | 0.90 | 0.78 |
| Figure 4D: Cortical neuron | $6.5e-9 \pm 4.2e-9$ | $0.59 \pm 0.16$ | 0.76 | 0.76 |

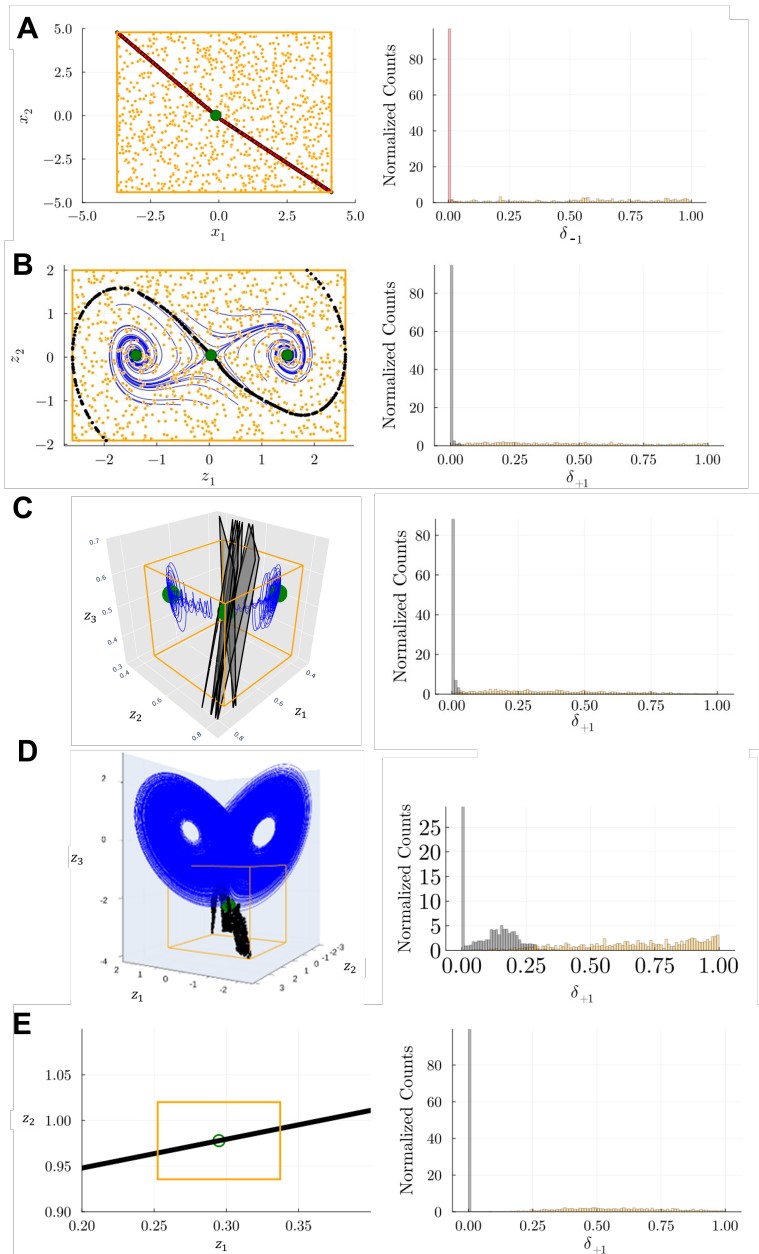

Figure 6: A) Left: Illustration of the sampling for Fig. 3A (left), with 1000 samples (red dots) drawn from the unstable manifold (reddish) and 1000 points (orange dots) drawn randomly from a box $\mathcal{U}$ as illustrated. Right: Histograms of $\delta_\sigma$ for $\boldsymbol{x}_0 \in W^{\{-1\}}(\boldsymbol{p}) \cap \mathcal{U}$ (red) and $\boldsymbol{x}_0 \in \mathcal{U}$ (orange). B) Left: Same as A for the bistable Duffing system from Fig. 4A, with points (black) sampled from the stable manifold (gray) vs. from an environment $\mathcal{U}$ (orange). Right: Histograms as in A. C) Left: Same as A for the choice model from Fig. 4B, with points (black) sampled from the stable manifold (gray) vs. from an environment $\mathcal{U}$ (orange). Right: Histograms as in A. D) Left: Same as A for the Loren63 system from Fig. 4C, with points (black) sampled from the stable manifold (gray) vs. from an environment $\mathcal{U}$ (orange). Right: Histograms as in A. E) Left: Same as A for the membrane potential recordings from Fig. 4D, with points (black) sampled from the stable manifold (gray) vs. from an environment $\mathcal{U}$ (orange). Note that this is a $2d$ projection of a 25-dimensional system, such that the orange box in this case only gives a rough idea of the sampling region surrounding the stable manifold. Right: Histograms as in A.

## H BENCHMARK SYSTEMS FOR TESTING

### H.1 ANALYTICAL TEST CASES

For our analytical test cases we chose example $2d$ PL maps from Gardini et al. (2009), which we can reformulate precisely as a PLRNN, such that a direct comparison with the ground truth is possible (i.e., without inference of the underlying DS as in the numerical examples from Fig. 4). In Gardini et al. (2009), the PL map considered has the $2d$ form

$$F(X) = \begin{cases} \boldsymbol{A_l} \cdot X + \boldsymbol{B}, & \text{if } x \leq 0, \\ \boldsymbol{A_r} \cdot X + \boldsymbol{B}, & \text{if } x \geq 0, \end{cases} \tag{48}$$

with $X = \begin{pmatrix} x \\ y \end{pmatrix}$ the two-coordinate state vector, and the transformation matrices and bias vector given by

$$\boldsymbol{A_l} = \begin{pmatrix} \tau_l & c \\ -\delta_l & d \end{pmatrix}, \quad \boldsymbol{A_r} = \begin{pmatrix} \tau_r & c \\ -\delta_r & d \end{pmatrix}, \quad \boldsymbol{B} = \begin{pmatrix} h_1 \\ h_2 \end{pmatrix}$$

This map can be reformulated as a PLRNN (Eq. 2) by defining the parameters $\boldsymbol{A}, \boldsymbol{W}, \boldsymbol{h}$ as:

$$\boldsymbol{A} = \begin{pmatrix} \tau_l & c \\ -\delta_l & d \end{pmatrix}, \quad \boldsymbol{W} = \begin{pmatrix} \tau_r - \tau_l & 0 \\ -\delta_r + \delta_l & 0 \end{pmatrix}, \quad \boldsymbol{h} = \begin{pmatrix} h_1 \\ h_2 \end{pmatrix}$$

For the analytical test case Fig. 3A, we chose parameters: (left) $\tau_r = 1.2$, $\delta_r = 1.8$, $\tau_l = -0.3$, $\delta_l = 0.9$, $c = -1.5$, $d = 1.0$, $h_2 = -0.1$, $h_1 = -0.13$, (right) $\tau_r = 1.26$, $\delta_r = 0.71$, $\tau_l = -0.39$, $\delta_l = -0.91$, $c = -0.44$, $d = 0.56$, $h_2 = 0.62$, $h_1 = -0.28$.

For the test case in Fig. 3B we adapt parameters from Figs. 3 & 4 in Gardini et al. (2009) as follows: (left) $\tau_r = -1.85$, $\delta_r = 0.9$, $\tau_l = -0.3$, $\delta_l = 0.9$, $c = 1$, $d = 0$, $h_2 = 0$, $h_1 = -1$, (right) $\tau_r = -1.85$, $\delta_r = 0.9$, $\tau_l = 0.9$, $\delta_l = 0.9$, $c = 1$, $d = 0$, $h_2 = 0$, $h_1 = -1$

For Fig. 5 we considered the following setting: $\tau_r = 1.5$, $\delta_r = 0.75$, $\tau_l = -1.77$, $\delta_l = 0.9$, $c = 0.6$, $d = 0.15$, $h_2 = -0.4$, $h_1 = -0.7$.

### H.2 RECURSIVE ALGORITHM FOR DETERMINING HOMOCLINIC INTERSECTIONS

Here we introduce another algorithm for finding homoclinic intersections in low-dimensional settings. We use this to validate our main algorithm in sect. 5. Specifically, to investigate the existence of homoclinic intersections for the map Eq. 48, we use an algorithm similar to the one proposed in (Roy et al., 2020). As illustrated in Fig. 7, it is based on a recursive procedure as follows:

1. Let $\mathcal{O}_{\mathcal{L}}^* = (x_{\mathcal{L}}^*, y_{\mathcal{L}}^*)^\mathsf{T} \in \mathcal{L}$ be an admissible saddle fixed point, and $\lambda_s$, $\lambda_u$ stable and unstable eigenvalues of $\boldsymbol{A_{\mathcal{L}}}$. Assume that $\boldsymbol{\ell}_s^*$ and $\boldsymbol{\ell}_u^*$ are the stable and unstable eigenlines generated by the associated stable and unstable eigenvectors, respectively. Analogous to Sect. I.2, suppose that $\boldsymbol{\ell}_u^*$ hits the border at $P_0 = (0, y_0)^\mathsf{T} \in \Sigma$ where $y_0$ is given by Eq. 60.

2. Consider the image of $P_0$ and assume this point is on the right side of the border, i.e. $P_1 \in \mathcal{S}$. Since this point has the coordinate $P_1 = (c\,y_0 + h_1, d\,y_0 + h_2)^\mathsf{T}$, it follows that

$$c\,y_0 + h_1$$
$$= c\,\frac{\big(b_l\,h_1 + (1 - a_l)h_2\big)(\lambda_u - d) - b_l\big(c\,h_2 + (1 - d)h_1\big)}{(\lambda_u - d)(1 - \Gamma_{\mathcal{L}} + D_{\mathcal{L}})}$$
$$+\, h_1 > 0. \tag{49}$$

Moreover $P_1 \in \ell^\Sigma$ is the first fold point of the unstable manifold of $\mathcal{O}_{\mathcal{L}}^*$, and so all its images will also be fold points.

3. Suppose that the orbit starting from $P_0$ will return to the border again. Let $n$ be the the border return time defined as the minimum number of iterations needed for $P_0$ to cross the

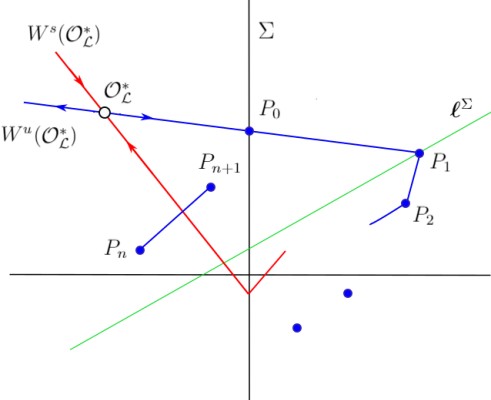

Figure 7: Schematic diagram to illustrate the procedure for finding homoclinic intersections.

border and return to the left side again, i.e., such that all iterations $P_1, P_2, \cdots, P_{n-1}$ lie on the right hand side while $P_n \in \mathcal{L}$. Using a recursive method similar to the one proposed in (Roy et al., 2020), we can compute the $n$-th iteration, $P_n$, directly from $P_0$. That is, there is no need to compute any other previous iterations $P_1, P_2, \cdots, P_{n-1}$. For this purpose, first we calculate $P_n$ as

$$P_n = T^n(P_0) = \boldsymbol{A}_{\mathcal{R}}^n P_0 + \left(\boldsymbol{A}_{\mathcal{R}} - \boldsymbol{I}\right)^{-1}\left(\boldsymbol{A}_{\mathcal{R}}^n - \boldsymbol{I}\right)\boldsymbol{h}. \tag{50}$$

Suppose that $\boldsymbol{A}_{\mathcal{R}}$ has two distinct eigenvalues, then, according to Proposition I.2, the matrix $\boldsymbol{A}_{\mathcal{R}}^n$ has the form Eq. 73. Hence, for $P_0 = (0, y_0)^{\mathsf{T}}$ we have

$$P_n = \begin{pmatrix} A_{n+1} - d\,A_n & c\,A_n \\ b_r A_n & d\,A_n - \mathcal{D}_{\mathcal{R}}\,A_{n-1} \end{pmatrix}\begin{pmatrix} 0 \\ y_0 \end{pmatrix} + \frac{-1}{\mathcal{P}_{\mathcal{R}}(1)} \times$$

$$\begin{pmatrix} 1-d & c \\ b_r & 1-a_r \end{pmatrix}\begin{pmatrix} A_{n+1} - dA_n - 1 & cA_n \\ b_r A_n & dA_n - \mathcal{D}_{\mathcal{R}}A_{n-1} - 1 \end{pmatrix}\begin{pmatrix} h_1 \\ h_2 \end{pmatrix} =$$

$$\begin{pmatrix} cA_n\,y_0 - \dfrac{\left((1-d)\left[A_{n+1}-dA_n-1\right]+cb_r A_n\right)h_1+c\left[A_n-\mathcal{D}_{\mathcal{R}}A_{n-1}-1\right]h_2}{\mathcal{P}_{\mathcal{R}}(1)} \\[4mm] \left(dA_n - \mathcal{D}_{\mathcal{R}}A_{n-1}\right)y_0 - \dfrac{b_r\left[A_n(1-\Gamma_{\mathcal{R}})+A_{n+1}-1\right]h_1+b^*\,h_2}{\mathcal{P}_{\mathcal{R}}(1)} \end{pmatrix} \tag{51}$$

where $A_n$ is given by Eq. 74; and $\Gamma_{\mathcal{R}}, \mathcal{D}_{\mathcal{R}}, \mathcal{P}_{\mathcal{R}}$ are the trace, determinant and characteristic polynomial of $\boldsymbol{A}_{\mathcal{R}}$ respectively; and

$$b^* = (a_r - 1)\left(1 + \mathcal{D}_{\mathcal{R}}A_{n-1}\right) + \left(d - \mathcal{D}_{\mathcal{R}}\right)A_n. \tag{52}$$

Now, by Eq. 51 we can compute all iterations and thus find the border return time $n$ for which the unstable manifold passes the border again.

4. Finally, we calculate $P_{n+1}$ and check whether or not the points $P_{n+1}$ and $P_n$ are on opposite sides of the stable eigenline $\ell_s^*$, i.e. whether we obtain a homoclinic intersection. For this, let $P_n = (x_n, y_n)^{\mathsf{T}}$, $P_{n+1} = (x_{n+1}, y_{n+1})^{\mathsf{T}}$, and consider $\mathsf{L}(x, y)$ given by Eq. 62. If $\mathsf{L}(x_n, y_n) \cdot \mathsf{L}(x_{n+1}, y_{n+1}) < 0$, then $P_n$ and $P_{n+1}$ are on opposite sides of the stable manifold, while $\mathsf{L}(x_n, y_n) \cdot \mathsf{L}(x_{n+1}, y_{n+1}) = 0$ implies at least one of the points $P_n$ or $P_{n+1}$ lies exactly on the stable manifold. In both cases there exists a homoclinic intersection.

---

**Algorithm 4** Investigating Homoclinic Intersections in 2D

---

1: **procedure** INVESTIGATEHOMOCLINICINTERSECTIONS
2:     $\mathcal{O}_{\mathcal{L}}^* \leftarrow (x_{\mathcal{L}}^*, y_{\mathcal{L}}^*)^{\mathsf{T}}$         ▷ Admissible saddle fixed point
3:     $(\lambda_s, \lambda_u) \leftarrow$ (Stable Eigenvalue, Unstable Eigenvalue)
4:     $(\ell_s^*, \ell_u^*) \leftarrow$ (Stable Eigenline, Unstable Eigenline)
5:     $P_0 \leftarrow (0, y_0)^{\mathsf{T}} \in \Sigma$       ▷ Unstable eigenline hits the border at $P_0$
6:     $P_1 \leftarrow (cy_0 + h_1, dy_0 + h_2)^{\mathsf{T}}$     ▷ Image of $P_0$ on the right side of the border
7:     $P_1 \in \ell^{\Sigma}$          ▷ First fold point of the unstable manifold
8:     $n \leftarrow$ Border return time    ▷ Minimum iterations for $P_0$ to return to the left side
9:     $P_n \leftarrow \boldsymbol{A}_{\mathcal{R}}^n P_0 + (\boldsymbol{A}_{\mathcal{R}} - \boldsymbol{I})^{-1}(\boldsymbol{A}_{\mathcal{R}}^n - \boldsymbol{I})\boldsymbol{h}$     ▷ Compute $P_n$ directly from $P_0$
10:     $P_{n+1} \leftarrow$ Next iteration      ▷ Check for homoclinic intersection
11:     **if** $\mathsf{L}(x_n, y_n) \cdot \mathsf{L}(x_{n+1}, y_{n+1}) < 0$ **then**
12:         **Return** "Homoclinic intersection exists"
13:     **else**
14:         **Return** "No homoclinic intersection"
15:     **end if**
16: **end procedure**

---

### H.3 SIMULATED BENCHMARKS

**Damped oscillator** The $2M$-dimensional damped oscillator, with $\boldsymbol{x}, \boldsymbol{y} \in \mathbb{R}^M$, is defined by

$$\dot{x}_i = y_i, \tag{53}$$

$$\dot{y}_i = -\alpha_i x_i - \beta_i x_i^3 - \gamma_i y_i \qquad i = 1, \ldots, M, \tag{54}$$

For the illustration in Fig. 2, we used $M = 10$ with parameters

$$\boldsymbol{\alpha} = [1.00, 0.80, 0.90, 0.30, 0.30, 0.25, 0.50, 0.89, 0.73, 0.21],$$

$$\boldsymbol{\beta} = [-0.02, -0.10, 0.01, -0.01, -0.011, 0.01, -0.06, -0.052, 0.03, -0.012],$$

$$\boldsymbol{\gamma} = [0.10, 0.05, 0.12, 0.07, 0.00, 0.03, 0.009, 0.04, 0.06, 0.0008].$$

**Duffing system** The *Duffing system* is a classical nonlinear DS that models a damped and periodically driven oscillator with a nonlinear restoring force. Originally introduced by Georg Duffing (Duffing, 1918), the system is described by a second-order differential equation of the form

$$\ddot{x} + \delta\dot{x} + \alpha x + \beta x^3 = \gamma \cos(\omega t),$$

where $x$ represents the displacement, $\delta$ is a damping coefficient, $\alpha$ and $\beta$ determine the linear and nonlinear stiffness, respectively, and $\gamma \cos(\omega t)$ is an external periodic forcing term. The Duffing system exhibits rich dynamical behavior, including periodic, quasi-periodic, and chaotic dynamics (depending on the forcing), making it a prototypical example in the study of nonlinear dynamics. Here we used for the parameters $\alpha = -1$, $\beta = 0.1$, $\delta = 0.5$, and $\gamma = 0$.

**Multistable decision making model** Simple models of decision making in the brain assume multistability between several choice-specific attractor states, to which the system's state is driven as one or the other choice materializes (Wang, 2002). Here we employ a simple multistable model from Gerstner et al. (2014), defined by

$$\tau_E \frac{dh_{E1}}{dt} = -h_{E1} + w_{EE}\, g_E(h_{E1}) + w_{EI}\, \gamma h + RI_1, \tag{55}$$

$$\tau_E \frac{dh_{E2}}{dt} = -h_{E2} + w_{EE}\, g_E(h_{E2}) + w_{EI}\, \gamma h + RI_2, \tag{56}$$

$$\tau_{\text{inh}} \frac{dh_{\text{inh}}}{dt} = -h_{\text{inh}} + w_{IE}(g_E(h_{E1}) + g_E(h_{E2})). \tag{57}$$

with

$$g_E(h; \theta) = \frac{1}{1 + e^{-(h-\theta)}}, \tag{58}$$

$$\tag{59}$$

and parameters:

$$\tau_E = 10, \qquad \tau_{\text{inh}} = 5, \qquad w_{EE} = 16, \qquad w_{EI} = -15, \qquad w_{IE} = 12,$$
$$R = 1, \qquad I_1 = 40, \qquad I_2 = 40, \qquad \gamma = 1 \qquad \theta = 5.$$

**Lorenz63** The *Lorenz63 system* (Lorenz, 1963) is a continuous-time dynamical system originally developed to model atmospheric convection. It describes the evolution of three state variables governed by the nonlinear differential equations

$$\frac{dx_1}{dt} = \sigma(x_2 - x_1),$$
$$\frac{dx_2}{dt} = x_1(\rho - x_3) - x_2,$$
$$\frac{dx_3}{dt} = x_1 x_2 - \beta x_3,$$

where $x_1$, $x_2$, and $x_3$ denote, respectively, the convection rate, horizontal temperature difference, and vertical temperature difference. The parameters $\sigma$, $\rho$, and $\beta$ correspond to physical constants related to the Prandtl number, Rayleigh number, and system geometry.

For specific values, e.g. $\sigma = 10$, $\rho = 28$, and $\beta = \frac{8}{3}$, the system exhibits chaotic dynamics. These settings give rise to the well-known "butterfly attractor," a prime example of deterministic chaos in low-dimensional systems.

### H.4 EMPIRICAL DATA (SINGLE CELL RECORDINGS)

For Fig. 4D, in-vitro membrane potential recordings from a single cortical neuron were used (Hertäg et al., 2012). Many types of cortical cells exhibit bistability between spiking activity and a stable equilibrium either near the resting potential or at a more depolarized level (Izhikevich, 2007; Durstewitz and Gabriel, 2007; Dovzhenok and Kuznetsov, 2012), and this example demonstrates how our algorithm can be utilized to reveal the structure of the state space supporting this type of dynamics from real cells.

# I   ADDITIONAL RESULTS

## I.1   ADDITIONAL FIGURES

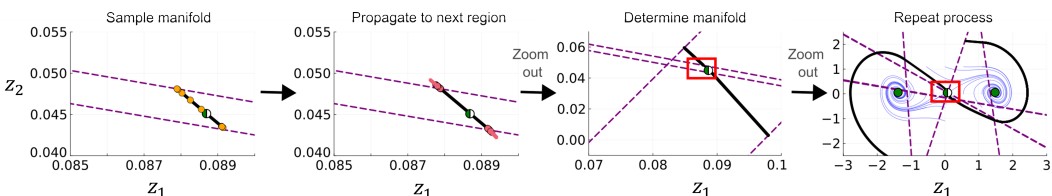

Figure 8: Illustration of the iterative procedure for computing stable manifolds with subregion boundaries of the shPLRNN ($M = 2$, $H = 10$) model in purple (dashed) Step 1: The stable manifold (black) is initialized using the stable eigenvector at the saddle point (half-green), and sample points (orange) are placed along it. Step 2: These points are propagated until they reach a new linear subregion, where the flow field is evaluated. Step 3: The updated manifold is given by the first principal component of points and the flow. Step 4: Repeating this process iteratively reconstructs the full global structure of the stable manifold (black), overlaid with the underlying GT flow field (blue)

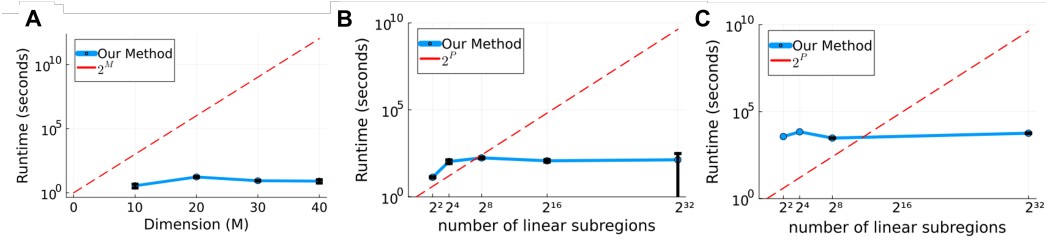

Figure 9: A) Algorithm runtime for determining the stable manifold of a saddle point in the Duffing system averaged across 5 runs (error bars = STDV). For a *constant* number of linear subregions $2^P$, Algo. 1's runtime hardly increases as a function of model size $M$ for an ALRNN ($P = 2$), confirming it is not significantly affected by the number of model parameters per se (note logarithmic scaling on y-axis). Algorithm 1 was run with constant $N_s = 1000$ and $N_{iter} = 2^P = 4$. Parameter $N_s$ was chosen deliberately high (much higher than usually required) to simulate the worst-case scenario. Runtime was determined on an Intel Core i5-1240P. B) For a *constant* model size $M$, Algo. 1's runtime was hardly affected by the number $2^P$ of linear subregions for an ALRNN ($M = 40$) when the manifold construction is restricted to the set of linear subregions explored by the data in the Duffing system. Algorithm 1 was run with $N_s = 1000$ and $N_{iter} = 2^P$. Runtime was determined on an Intel Core i5-1240P. C) Algorithm runtime for determining the stable manifold of a saddle point averaged across 3 runs (error bars = STDV) for the Lorenz63. For a *constant* model size $M$, Algo. 1's runtime was hardly affected by the number $2^P$ of linear subregions for an ALRNN ($M = 100$) when the manifold construction is restricted to the set of linear subregions explored by the data. The manifold was computed with $N_s = 5000$ and $N_{iter} = 2^P$; as in A-B, $N_s$ was chosen deliberately high to cover the worst case. All models were trained on the Lorenz system. Runtime was determined on an Intel Xeon Gold 6248. We emphasize that in general the scaling may strongly depend on the system's actual dynamics and topological structure, such that general statements regarding scaling are therefore difficult.

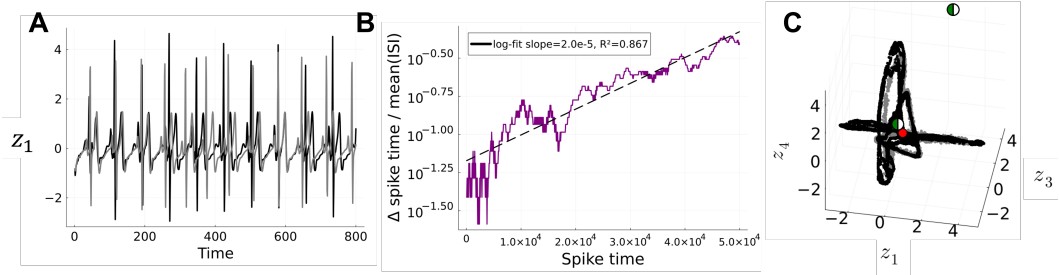

Figure 10: ALRNN ($M = 80$, $P = 4$) trained on a 5-dimensional embedding of a human ECG recording. A) Example of ALRNN-simulated ECG activity (black) and true human ECG (gray). B) Exponential growth in differences in spike timing between ALRNN-generated trajectories from two slightly different initial conditions as a function of spike time (ISI = interspike interval). While the divergence is exponential (note the logarithmic scale of the y-axis), thus suggesting the presence of chaos, the divergence rate appears to be rather small. This impression is corroborated by the estimated slightly positive maximum Lyapunov exponent of $0.0074 \pm 0.00083$ (mean ± SD), while the sum of all Lyapunov exponents is $< 0$, as estimated from the model Jacobians evolved for 10,000 time steps for 5 different initial conditions. It thus appears the system is weakly chaotic (Alligood et al., 1996; Guckenheimer and Holmes, 2013), but it is difficult to conclusively establish this from these observations alone. C) Visualization of the presumably chaotic attractor for the real data (gray, via delay-embedding; Kantz and Schreiber (2004)) and as generated by the ALRNN (black) in a $3d$ subspace of the 80-dimensional state space. A heteroclinic intersection (within the limits of numerical precision, $\approx 10^{-10}$) of the saddle nodes in green was identified by our algorithm at the red dot, thus confirming the presence of chaos in this system.

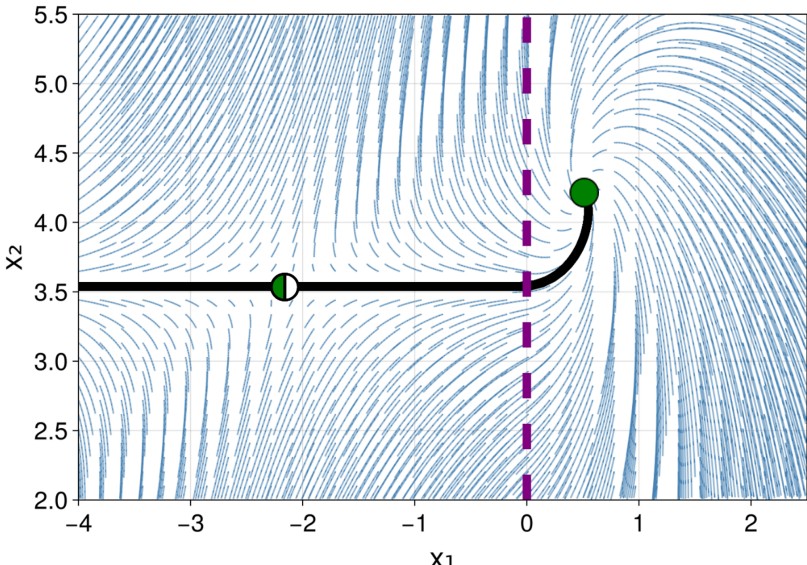

Figure 11: Illustration of how curvature can arise in the un-/stable manifolds as borders to other linear subregions are crossed. The stable manifold (black) of the saddle (half-green) is a straight line in the first linear subregion (left hand side), but becomes a curve as it enters the second subregion (right from dashed purple border), where the dynamics follows a spiral. The parameters of the $2d$ PLRNN used for illustration were: $A = [0.93\ 0.0; 0.0\ 0.92]$, $W = [0.26\ 0.080; -0.21\ 0.24]$, $h = [-0.43,\ -0.57]$.

### I.2 ANALYTICAL CONDITION FOR HOMOCLINIC INTERSECTION OF THE FIRST AND SECOND FOLD POINTS

Homo- or heteroclinic intersections between stable and unstable manifolds imply the existence of infinitely many such intersections, leading to a 'horseshoe structure' and the existence of chaotic orbits (Wiggins, 1988). In this subsection we derive the analytical results needed to confirm such intersections using Algorithm 4.

Let $\mathcal{O}_{\mathcal{L}}^* = (x_{\mathcal{L}}^*, y_{\mathcal{L}}^*)^\mathsf{T}$ be an admissible saddle fixed point in the left sub-region $\mathcal{L}$. Since $\mathcal{O}_{\mathcal{L}}^*$ is a saddle, $\boldsymbol{A}_{\mathcal{L}}$ has one stable and one unstable eigenvalue $\lambda_s$ and $\lambda_u$, respectively. Let us denote the line generated by the associated stable eigenvector by $\boldsymbol{\ell}_s^*$ and the line produced by the corresponding unstable eigenvector by $\boldsymbol{\ell}_u^*$. Since the unstable eigenvector is $\boldsymbol{v}_u = (v_1, v_2)^\mathsf{T} = (\frac{\lambda_u - d}{b_l}, 1)^\mathsf{T}$, $\boldsymbol{\ell}_u^*$ hits the border $x = 0$ at $P_0 = (0, y_0)^\mathsf{T} \in \Sigma$ where

$$
\begin{aligned}
y_0 &= y_{\mathcal{L}}^* - x_{\mathcal{L}}^* \frac{v_2}{v_1} \\
&= \frac{b_l h_1 + (1 - a_l) h_2}{1 - \Gamma_{\mathcal{L}} + D_{\mathcal{L}}} - \frac{(1 - d) h_1 + c h_2}{1 - \Gamma_{\mathcal{L}} + D_{\mathcal{L}}} \left( \frac{b_l}{\lambda_u - d} \right) \\
&= \frac{\big(b_l h_1 + (1 - a_l) h_2\big)(\lambda_u - d) - b_l\big(c h_2 + (1 - d) h_1\big)}{(\lambda_u - d)(1 - \Gamma_{\mathcal{L}} + D_{\mathcal{L}})}.
\end{aligned}
\tag{60}
$$

The image of $P_0$ is the first fold point of the unstable manifold of $\mathcal{O}_{\mathcal{L}}^*$, and so all its images will also be fold points. Its coordinate is $P_1 = (c y_0 + h_1, d y_0 + h_2)^\mathsf{T}$. The image of the first fold point is the second fold point $P_2 = (x_2, y_2)^\mathsf{T}$ with coordinates

$$
\begin{cases}
x_2 = c(a_l + d)y_0 + (a_l + 1)h_1 + ch_2 \\
y_2 = (b_l c + d^2)y_0 + b_l h_1 + (d + 1)h_2
\end{cases}
\quad, \quad \text{if } c\,y_0 + h_1 < 0
$$

$$
\begin{cases}
x_2 = c(a_r + d)y_0 + (a_r + 1)h_1 + ch_2 \\
y_2 = (b_r c + d^2)y_0 + b_r h_1 + (d + 1)h_2
\end{cases}
\quad, \quad \text{if } c\,y_0 + h_1 > 0
\tag{61}
$$

Now we check whether or not the points $P_1$ and $P_2$ are on opposite sides of the stable eigenline $\boldsymbol{\ell}_s^*$. When $P_1$ and $P_2$ are on opposite sides of $\boldsymbol{\ell}_s^*$, then the unstable manifold must have intersected the stable manifold. Thus, we have a homoclinic intersection which implies the occurrence of chaotic dynamics. Since the stable eigenvector is $\boldsymbol{v}_s = (\frac{\lambda_s - d}{b_l}, 1)^\mathsf{T}$ and the eigenline $\boldsymbol{\ell}_s^*$ passes through $\mathcal{O}_{\mathcal{L}}^* = (x_{\mathcal{L}}^*, y_{\mathcal{L}}^*)^\mathsf{T}$, $\boldsymbol{\ell}_s^*$ can be computed as

$$
\begin{aligned}
\boldsymbol{\ell}_s^*: \quad &\frac{\lambda_s - d}{b_l} y - x + \frac{(1 - d) h_1 + c h_2}{1 - \Gamma_{\mathcal{L}} + D_{\mathcal{L}}} \\
&- \left( \frac{\lambda_s - d}{b_l} \right) \frac{b_l h_1 + (1 - a_l) h_2}{1 - \Gamma_{\mathcal{L}} + D_{\mathcal{L}}} =: \mathsf{L}(x, y) = 0.
\end{aligned}
\tag{62}
$$

Now there are two possibilities:

**Case I**: $\mathsf{L}(x_1, y_1) \cdot \mathsf{L}(x_2, y_2) \leq 0$

If $\mathsf{L}(x_1, y_1) \cdot \mathsf{L}(x_2, y_2) < 0$, then $P_1$ and $P_2$ are on opposite sides of the stable manifold, while $\mathsf{L}(x_1, y_1) \cdot \mathsf{L}(x_2, y_2) = 0$ implies that at least one of the points $P_1$ and $P_2$ lies exactly on the stable manifold. In both cases there exists a homoclinic intersection, i.e., whenever we have

$$
\begin{aligned}
&\left( \frac{\lambda_s - d}{b_l}(dy_0 + h_2) - (cy_0 + h_1) + \mathcal{M} \right)\left( \frac{\lambda_s - d}{b_l}\big((b_{l/r}c + d^2)y_0 \right. \\
&\left. + b_{l/r}h_1 + (d + 1)h_2\big) - \big(c(a_{l/r} + d)y_0 + (a_{l/r} + 1)h_1 + ch_2\big) \right. \\
&\left. + \mathcal{M} \right) \leq 0,
\end{aligned}
\tag{63}
$$

where $\mathcal{M} = \frac{(1-d)\,h_1 + c\,h_2}{1 - \Gamma_{\mathcal{L}} + D_{\mathcal{L}}} - \left(\frac{\lambda_s - d}{b_l}\right) \frac{b_l\,h_1 + (1-a_l)\,h_2}{1 - \Gamma_{\mathcal{L}} + D_{\mathcal{L}}}$.

The homoclinic intersection point $P_{hom} = (x_{hom}, y_{hom})^{\mathsf{T}}$ is the point of intersection between the line joining the two fold points and the stable eigenline $\ell_s^*$, and hence given by

$$\begin{cases} x_{hom} = (x_2 - x_1)\beta + x_1 \\ y_{hom} = (y_2 - y_1)\beta + y_1 \end{cases}, \tag{64}$$

where

$$\beta = \frac{x_1 - \frac{\lambda_s - d}{b_l} - \mathcal{M}}{\frac{\lambda_s - d}{b_l}(y_2 - y_1) - (x_2 - x_1)}. \tag{65}$$

Finally, we must ensure that the intersection happens before the stable eigenline hits the border. For this, we need to have $x_{home}\,x_{\mathcal{L}}^* > 0$, which implies the intersection point $P_{hom}$ and the fixed point $\mathcal{O}_{\mathcal{L}}^*$ are on the same side of $\Sigma$.

**Case II**: $\mathsf{L}(x_1, y_1) \cdot \mathsf{L}(x_2, y_2) > 0$

If $\mathsf{L}(x_1, y_1) \cdot \mathsf{L}(x_2, y_2) > 0$, we need to check whether the unstable manifold intersects with the part of the stable manifold which ensues after folding along the $y$-axis. For this we have to calculate more points of the global stable manifold. Since $T$ is assumed to be invertible, the global stable manifold is formed by the union of all preimages (inverses) of any rank of the local stable set (a segment of the local stable eigenline). Assume, under the action of $T^{-1}$, the line $\ell_s^*$ maps to the $y$-axis and intersects it at $\tilde{P}_0 = (0, \tilde{y}_0)^{\mathsf{T}} \in \Sigma$. Then $\tilde{P}_0$ is the first fold point of the stable manifold of $\mathcal{O}_{\mathcal{L}}^*$, and $\tilde{y}_0$ is given by

$$\tilde{y}_0 = \frac{\big(b_l\,h_1 + (1 - a_l)h_2\big)(\lambda_s - d) - b_l\big(c\,h_2 + (1 - d)h_1\big)}{(\lambda_s - d)(1 - \Gamma_{\mathcal{L}} + D_{\mathcal{L}})}. \tag{66}$$

The preimage of $\tilde{P}_0$ is the second fold point $\tilde{P}_{-1} = (\tilde{x}_{-1}, \tilde{y}_{-1})^{\mathsf{T}}$ with coordinates

$$\begin{cases} \tilde{x}_{-1} = \frac{1}{D_{\mathcal{L}}}\big(-c\tilde{y}_0 + ch_2 - dh_1\big) \\ \tilde{y}_{-1} = \frac{1}{D_{\mathcal{L}}}\big(a_l\tilde{y}_0 + b_lh_1 - a_lh_2\big) \end{cases}, \quad \text{if } \frac{\varphi^{\mathsf{T}}(\tilde{P}_0 - \boldsymbol{h})}{D_{\mathcal{L}}} \leq 0$$

$$\begin{cases} \tilde{x}_{-1} = \frac{1}{D_{\mathcal{R}}}\big(-c\tilde{y}_0 + ch_2 - dh_1\big) \\ \tilde{y}_{-1} = \frac{1}{D_{\mathcal{R}}}\big(a_r\tilde{y}_0 + b_rh_1 - a_rh_2\big) \end{cases}, \quad \text{if } \frac{\varphi^{\mathsf{T}}(\tilde{P}_0 - \boldsymbol{h})}{D_{\mathcal{R}}} \geq 0 \tag{67}$$

where

$$\frac{\varphi^{\mathsf{T}}(\tilde{P}_0 - \boldsymbol{h})}{D_{\mathcal{L}/\mathcal{R}}} = \frac{-d\,h_1}{D_{\mathcal{L}/\mathcal{R}}} - \frac{c}{D_{\mathcal{L}/\mathcal{R}}}(\tilde{y}_0 - h_2). \tag{68}$$

Now the line joining the two fold points $\tilde{P}_0$ and $\tilde{P}_{-1}$ is given by $\tilde{\mathsf{L}}(x, y) = 0$ where

$$\tilde{\mathsf{L}}(x, y) := y - \tilde{y}_0 - \frac{\tilde{y}_{-1} - \tilde{y}_0}{\tilde{x}_{-1}} x. \tag{69}$$

If $\tilde{\mathsf{L}}(x_1, y_1) \cdot \tilde{\mathsf{L}}(x_2, y_2) < 0$, then $P_1$ and $P_2$ are on opposite sides of the stable manifold, and the unstable manifold intersects the stable manifold at $\tilde{P}_{hom} = (\tilde{x}_{hom}, \tilde{y}_{hom})^{\mathsf{T}}$ with

$$\begin{cases} \tilde{x}_{hom} = (x_2 - x_1)\tilde{\beta} + x_1 \\ \tilde{y}_{hom} = (y_2 - y_1)\tilde{\beta} + y_1 \end{cases}, \tag{70}$$

where

$$\tilde{\beta} = \frac{(\tilde{y}_0 - y_1)\tilde{x}_{-1} + (\tilde{y}_{-1} - \tilde{y}_0)x_1}{(y_2 - y_1)\tilde{x}_{-1} - (\tilde{y}_{-1} - \tilde{y}_0)(x_2 - x_1)}. \tag{71}$$

We need to have $\tilde{x}_{home}\,\tilde{x}_{-1} > 0$, which means the intersection point $\tilde{P}_{hom}$ and the point $\tilde{P}_{-1}$ are on the same side of $\Sigma$.

*Remark* I.1. An analogous procedure can be performed to analytically obtain homoclinic intersections for the fixed point $\mathcal{O}_{\mathcal{R}}^* \in \mathcal{R}$.

### I.2.1 HOMOCLINIC INTERSECTIONS FOR FURTHER ITERATIONS OF FOLD POINTS

In order to determine homoclinic intersections, it may be necessary to check further iterations of fold points for which we would need a recursive procedure. This will involve calculating the $n$-th power of a $2 \times 2$ matrix, and we therefore first prove the following Proposition:

**Proposition I.2.** *Let* $M = \begin{pmatrix} a & c \\ b & d \end{pmatrix}$ *have two distinct eigenvalues*

$$\lambda_{1,2} = \frac{a+d}{2} \mp \frac{\sqrt{(a-d)^2 + 4\,b\,c}}{2} = \frac{\Gamma}{2} \mp \frac{\sqrt{\Gamma^2 - 4\,\mathcal{D}}}{2}, \tag{72}$$

*where* $\Gamma$ *and* $\mathcal{D}$ *are the trace and determinant of* $M$. *Then, for every* $n \in \mathbb{N}$

$$M^n = \begin{pmatrix} A_{n+1} - d\,A_n & c\,A_n \\ b A_n & d\,A_n - \mathcal{D}\,A_{n-1} \end{pmatrix} \tag{73}$$

*where*

$$A_n = \frac{\lambda_1^n - \lambda_2^n}{\lambda_1 - \lambda_2}. \tag{74}$$

*Proof.* In our case $M$ is diagonalizable, so for $b \neq 0$

$$M = V\,\Lambda\,V^{-1}$$

$$= \frac{1}{\lambda_1 - \lambda_2} \begin{pmatrix} \frac{\lambda_1 - d}{b} & \frac{\lambda_2 - d}{b} \\ 1 & 1 \end{pmatrix} \begin{pmatrix} \lambda_1 & 0 \\ 0 & \lambda_2 \end{pmatrix} \begin{pmatrix} b & d - \lambda_2 \\ -b & \lambda_1 - d \end{pmatrix}. \tag{75}$$

Therefore

$$M^n = V\,\Lambda^n\,V^{-1}$$

$$= \frac{1}{\lambda_1 - \lambda_2} \begin{pmatrix} \frac{\lambda_1 - d}{b} & \frac{\lambda_2 - d}{b} \\ 1 & 1 \end{pmatrix} \begin{pmatrix} \lambda_1^n & 0 \\ 0 & \lambda_2^n \end{pmatrix} \begin{pmatrix} b & d - \lambda_2 \\ -b & \lambda_1 - d \end{pmatrix}$$

$$= \frac{1}{\lambda_1 - \lambda_2} \times$$

$$\begin{pmatrix} \lambda_2^n(d - \lambda_2) - \lambda_1^n(d - \lambda_1) & -\frac{(d - \lambda_1)(d - \lambda_2)}{b}(\lambda_1^n - \lambda_2^n) \\ b(\lambda_1^n - \lambda_2^n) & \lambda_1^n(d - \lambda_2) - \lambda_2^n(d - \lambda_1) \end{pmatrix}$$

$$= \begin{pmatrix} A_{n+1} - d\,A_n & -\frac{d^2 - \Gamma\,d + \mathcal{D}}{b}\,A_n \\ b A_n & d\,A_n - \mathcal{D}\,A_{n-1} \end{pmatrix}$$

$$= \begin{pmatrix} A_{n+1} - d\,A_n & c\,A_n \\ b A_n & d\,A_n - \mathcal{D}\,A_{n-1} \end{pmatrix}. \tag{76}$$

If $b = 0$, then the eigenvalues of $M$ are $\lambda_1 = a$ and $\lambda_2 = d$ ($a \neq d$). Thus,

$$M = V\,\Lambda\,V^{-1} = \begin{pmatrix} 1 & \frac{-c}{a-d} \\ 0 & 1 \end{pmatrix} \begin{pmatrix} a & 0 \\ 0 & d \end{pmatrix} \begin{pmatrix} 1 & \frac{c}{a-d} \\ 0 & 1 \end{pmatrix}, \tag{77}$$

and so (for $n \in \mathbb{N}$)

$$M^n = V\,\Lambda^n\,V^{-1} = \begin{pmatrix} 1 & \frac{-c}{a-d} \\ 0 & 1 \end{pmatrix} \begin{pmatrix} a^n & 0 \\ 0 & d^n \end{pmatrix} \begin{pmatrix} 1 & \frac{c}{a-d} \\ 0 & 1 \end{pmatrix}$$

$$= \begin{pmatrix} a^n & \frac{c(a^n - d^n)}{a-d} \\ 0 & d^n \end{pmatrix}, \tag{78}$$

which yields equation Eq. 73 for $b = 0$. $\qquad \square$

