# OpenReview forum: "Detecting Invariant Manifolds in ReLU-Based RNNs"
_ICLR.cc/2026/Conference — ICLR 2026 Poster_

### Official Review · Reviewer_KgUd · 2025-10-29

**Soundness:** 3
**Presentation:** 3
**Contribution:** 3
**Rating:** 6
**Confidence:** 4

**Summary:**

This paper presents a new algorithm for identifying stable and unstable manifolds in piecewise-linear recurrent neural networks (PLRNNs). The authors leverage the linearity within each ReLU activation region to derive an efficient, semi-analytical method that constructs manifolds by iteratively propagating sampled points across subregions. This approach enables explicit detection of manifolds that delineate basins of attraction and reveal multistability or chaotic structures in RNN dynamics. The authors demonstrate the method on several systems including the Duffing oscillator, Lorenz-63 attractor, decision-making RNNs, and real neuronal recordings. The results show close agreement between the algorithmic manifolds and those computed analytically or numerically from the underlying dynamical systems, suggesting both accuracy and scalability beyond traditional continuation techniques.

**Strengths:**

The paper is clearly written and well-organized. The proposed algorithm effectively leverages the piecewise-linear structure of PLRNNs, leading to an elegant and computationally efficient solution. The authors successfully ground their work in established theoretical principles from dynamical systems theory and neuroscience, making a strong case for its relevance to both machine learning and scientific modeling. The experimental section is broad and persuasive, spanning toy systems, classical chaotic dynamics, and real neural recordings, which together illustrate the versatility of the method and its capacity to illuminate the internal mechanisms of learned recurrent dynamics. The addition of invertibility regularization and supporting theoretical analysis further reflects the authors’ careful consideration of the method’s stability and practical applicability.

**Weaknesses:**

As far as I can tell, the paper does not engage with the existing literature on Threshold-Linear Networks (TLNs), particularly the extensive work by Carina Curto and collaborators. This omission is notable, as TLNs share strong conceptual and mathematical parallels with the proposed framework. The TLN literature provides rigorous theoretical results on how ReLU-based recurrent networks partition state space into distinct regions, each exhibiting its own dynamical behavior—precisely the type of structure this paper seeks to characterize.

For instance, see [this paper](https://arxiv.org/pdf/2109.03198) and [this paper](https://arxiv.org/pdf/1605.04463).

**Questions:**

Can the authors please clarify the relationship of PLRNNs to TLNs, and situate their findings within this literature?

---

> ### Author Response · Authors · 2025-11-20
> **Rebuttal**
>
> Thank you very much for this very encouraging and positive assessment of our work!
>
> **Weaknesses**
>
> Yes, Curto and her group provided a lot of amazing and beautiful mathematical results about TLNs [1-3], which are indeed closely related to PLRNNs. More generally, apart from TLNs, there are other PL systems like, for instance, switching linear DS [4,5], that have a similar structure, and to which our manifold algorithm may in principle be applicable as well. We brought this up now both in the Introduction (lines 83-91), as well as in the Conclusions on the last page, where we now included a new pg. connecting our developments to the existing literature on PL models in other areas, including TLNs and the work by Curto & colleagues. **The new additions are all highlighted in blue text color.**
>
> **Questions**
>
> Specifically, TLNs are a kind of a continuous-time version of the shPLRNN [6], which we also consider as one of our models. However, the fact that TLNs are defined in continuous time (as systems of ODEs) and PLRNNs in discrete time (as recursive maps, like most RNN models in ML/AI) implies that we cannot directly transfer all theoretical results from the TLN to the PLRNN literature (as the behavior of maps can be different from their continuous-time counterparts, e.g. [7]). Moreover and more importantly, in much of the theoretical literature on TLNs, to which Curto and colleagues have fundamentally contributed, specific restrictions are imposed on the structure of the weight matrices, like binarized forms or inhibition-dominated; these restrictions enable much of the theoretical analysis in the first place (as far as we are aware of). In contrast, in the PLRNN literature [6,9,10], weight matrices $W \in \mathbb{R}^{M \times M}$ are fully trainable, without any restrictions, which is indeed essential to train these models on DS reconstruction tasks. Thus, much of the theoretical analysis in the mentioned papers is unfortunately not directly applicable to our setting, as far as we can see. More specifically, the computation of un-/stable manifolds, the focus of our work, has not been considered in the TLN literature as far as we are aware of.
>
>
> **References:**
>
> [1] C. Curto and K. Morrison. Graph rules for recurrent neural network dynamics. Notices of the American Mathematical Society, 70(04), 2023.
>
> [2] K. Morrison, A. Degeratu, V. Itskov, and C.Curto. Diversity of emergent dynamics in competitive threshold-linear networks. arXiv preprint arXiv:1605.04463, 2016
>
> [3]C. Parmelee, S.a Moore, K. Morrison, and C. Curto. Core motifs predict dynamic attractors in combinatorial threshold-linear networks. PloS one, 17(3):e0264456, 2022
>
> [4]  S. W. Linderman and M. J. Johnson. Structure-Exploiting variational inference for recurrent switching linear dynamical systems. In 2017 IEEE 7th International Workshop on Computational Advances in Multi-Sensor Adaptive Processing (CAMSAP), pages 1–5, December 2017
>
> [5] S. W. Linderman, A. C. Miller, R. P. Adams, D. M. Blei, L. Paninski, and M. J. Johnson. Recurrent switching linear dynamical systems. arXiv:1610.08466 [stat], October 2016.
>
> [6] F. Hess, Z. Monfared, M. Brenner, and D. Durstewitz. Generalized teacher forcing for learning chaotic dynamics. In Proceedings of the 40th International Conference on Machine Learning, volume 202 of Proceedings of Machine Learning Research, pages 13017–13049. PMLR, 23–29 Jul 2023
>
> [7] J. Guckenheimer, & P. Holmes. (2013). Nonlinear oscillations, dynamical systems, and bifurcations of vector fields (Vol. 42). Springer Science & Business Media.
>
> [8] R. Devaney (2018). An introduction to chaotic dynamical systems. CRC press.
>
> [9] M. Brenner, C. J. Hemmer, Z. Monfared, and D. Durstewitz. Almost-linear rnns yield highly interpretable symbolic codes in dynamical systems reconstruction. In A. Globerson, L. Mackey, D. Belgrave, A. Fan, U. Paquet, J. Tomczak, and C. Zhang, editors, Advances in Neural Information Processing Systems, volume 37, pages 36829–36868. Curran Associates, Inc., 2024a
>
> [10]. Brenner, F. Hess, J. M. Mikhaeil, L. F. Bereska, Z. Monfared, P. Kuo, and D. Durstewitz. Tractable Dendritic RNNs for Reconstructing Nonlinear Dynamical Systems. In Proceedings of the 39th International Conference on Machine Learning, pages 2292–2320. PMLR, June 2022

---

### Official Review · Reviewer_MbXp · 2025-10-31

**Soundness:** 2
**Presentation:** 3
**Contribution:** 2
**Rating:** 4
**Confidence:** 3

**Summary:**

This paper discusses the problem of finding stable and unstable manifolds in a class of recurrent neural networks. The authors treat RNN's involving the ReLU activation function which allows them to treat the system as a piecewise linear system. From what I understand of the paper, the authors exploit the piecewise linear property of the system and use the eigendecomposition of the Jacobian as the basis for computing the manifolds. They have performed some tests on various systems, some simple, some more complex and their algorithm appears to work well.

**Strengths:**

I found the paper interesting and relevant. I have read numerous papers on computing stable/unstable manifolds in nonlinear systems, but these have typically been in continuous time and I think this paper is one of the few (or perhaps only) paper which treats this specific type of system. It should be observed that generally the computation of stable/unstable manifolds for nonlinear systems is a very hard problem and one which gets harder as the order of the system increases and the dynamics become more complex.

The writing in the paper was good in terms of well constructed prose and a well-presented manuscript.

The authors' algorithm appeared to work well on the examples tested.

**Weaknesses:**

My chief complaint about the paper is it meandered awkwardly between giving the reader quite basic details (and perhaps too much of these) in some places and then depriving the reader of significant details of the authors' contribution. In some places it was not clear what was obvious, what was the authors' work and what was new. The inclusion of a few more references at particular places would be useful. The paper is quite easy to understand up until Section 3.3 when I felt the authors needed to give significantly more detail. In particular I thought:-

1. The authors should give more detail about Algorithm 1, which does not appear to have been described comprehensively. Indeed, it does not seem to referenced in the main text until Section 4. I would like to see a better, more comprehensive explanation of the ideas behind it, its construction, and a more detailed description of its steps and the various mathematical objects it contains. For example in Algorithm 1 $\sigma$ is an index, but what values does it take (one can guess? One expects $Q$ to be the manifold, but in the algorithm it seems like $Q^{\sigma}_k$ is a selection of points in that manifold? I am not sure. Also in Step 4 it seems like $n$ denotes iteration number, but is this actually $k$? These issues are probably resolvable, but without some sort of description, make the paper only possible to appreciate at a superficial level.

2. I was not entirely sure how the SCYFI package was used. Does this provide the points $P$ for Algorithm 1? Please explain and perhaps also indicate how $N_{max}$ is chosen.

3. In Section 3.4 the authors discuss regularization which certainly makes sense. However, the details of how this regularizing term is included is not clear: to which "loss" is the term (6) included? In the discussion below equation (6), the authors refer to Figure 3 (top), (bottom) and (right), but this does not seem to correspond with the figures included in the paper - there is only one row of figures in Figure 3. I was confused.

4. I found the appendices quite long and strangely structured. It was not quite clear to me how central some of these were to the paper. For example the first few paragraphs of Appendix D seemed to be only weakly relevant to the paper. Section D.1 seemed relevant, but then Section D.2 seemed rather verbose and un-focused. The relevance of the appendices to the main text needs more focus. Finally, it seems that some sort of "training" is needed for the algorithm reported, according to the appendix, but this does not seem to be discussed in the main text. Some clarity here would be useful for the reader.

5. Finally a comment about the literature. The system which the authors consider belong to a type of so-called "Lur'e" or "Lurie" or "Lurye" systems considered in the control theory literature. Furthermore the PLL RNN which they consider seems quite close to the piecewise linear systems again considered in the control theory literture. I am not asking the authors to cite any specific papers, but they may find it useful to consult some of this literature.

**Questions:**

The examples look good, but to me it was not clear exactly what the algorithm returned when computing manifolds - was it just samples from the manifold and then one would deduce from these samples the likely shape of the manifold? This would of course be dependent on the density of samples used, particularly in critical regions of the state-space. Some more explanation and guidance would be useful.

---

> ### Author Response · Authors · 2025-11-20
> **Rebuttal #1 (Weaknesses)**
>
> We thank the referee for the constructive and insightful comments, which led us to rework major parts of our manuscript. **All text changes are highlighted in blue in the revised version we have uploaded.**
>
> **Weaknesses**
>
> **W1 (details of algorithm unclear/ missing):** We completely rewrote sect. 3.3 and Algorithm 1 in order to clarify the precise steps, assumptions, numerical vs. analytical aspects, and mathematical objects; please check our revised version as uploaded for details. It should also be clear now from sect. 3.3 (with appropriate references) what we have used from previous work (essentially only the SCYFI algorithm for locating the saddle points) and what is genuinely our own and novel contribution (the complete manifold algorithm as summarized in Algorithm 1, as well as the regularization introduced in sect. 3.4 as required in this context). We have also removed in sect. 3.2 some details that may have been too basic.
>
> In brief, with regards to your specific questions: $\sigma \in \\{-1,+1\\}$ is simply a flag indicating the type of manifold to be computed (-1=unstable, +1=stable). $Q$ stores the support vectors which parameterize the manifold; these will be the eigenvectors of the PCA or kernel-PCA which define the orientation of the manifold segments within each linear subregion $k$, and one support point which anchors the manifold segment in the space. Since in each iteration $n$ more than just one neighboring subregion may be visited, $k \neq n$ in general (but we have changed this notation now in Algorithm 1 for clarity, and presented the steps in more detail).
>
> **W2 (usage of SCYFI):** Yes, we use SCYFI to first detect all cyclic points for which we would like to compute the un-/stable manifolds (now clarified). $N_{max}$ (changed to $N_{iter}$ for clarity) in Algorithm 1 is a hyperparameter which determines the maximum number of iterations through subregions. This may simply be set to the total number of subregions; the algorithm will automatically terminate once no new regions are reached, but sometimes it may be more practical to focus on a subset of relevant subregions which contain the data of interest. This is now also clarified in the rewrite of sect. 3.3 and in Algorithm 1.
>
> **W3 (regularization loss & Fig. reference):** Thanks for pointing out and apologies for the confusion, this should have been Fig. 2 of course (now fixed)! The regularization term is added to the loss of the RNN to be trained, in our case a conventional MSE loss (Eq. 40 in Appx. E), but in theory this could be any other type of reconstruction loss. We have now clarified this in sect. 3.4 and added more details on RNN training to Appx. E.
>
> **W4 (structure of appendices):** We fully agree. The Appx. contained a lot of material that we had assembled in preparation for the main studies and found interesting in this context, but which is not necessary to understand the algorithm we develop in the main text or the examples shown there. We have thus now removed many sections which were only loosely related, clarified the relevance of the remaining sections, and reworked and reorganized the whole Appx. such that different subsections now better reflect thematically related material and its relevance to the main paper.
>
> Regarding training: No, the manifold algorithm itself does not require any training, it is a semi-analytical construction (as should be clear now from rewritten sect. 3.3 and Algorithm 1).
> However, to learn a dynamical system (DS) from time series data like those for the systems in Fig. 4 (see Appx. H.3), we first need to train a DS reconstruction model, in our case different variants of piecewise-linear RNNs (sect. 3.1 \& Appx. E), on time series data from the system. These models themselves, and training algorithms for them, are well established and are not part of the methodological developments in the present paper, as we now clarified (see sect. 3.1, 3.4, and Appx. E).
>
> **W5 (additional literature):** Thank you for pointing this out, absolutely right, Lur’e systems are indeed closely related to the PLRNNs we are considering here (but usually defined in continuous, not in discrete time). We included references to this body of literature now on lines 83-91 of the Introduction, and - more generally - extended our discussion of related PL systems as they have been used in mathematics and engineering both in sect. 1 and in the Conclusions, sect. 6 (as highlighted in blue).

---

> > ### Author Response · Authors · 2025-11-20
> > **Rebuttal #2 (Questions)**
> >
> > **Questions (output of algorithm)**
> >
> > We clarified this now in our rewrite of sect. 3.3 \& Algorithm 1, see comment on W1 above. In brief, our algorithm provides a complete analytical specification of the manifolds, not just some points sampled from it. For this complete specification, we only need a small set of support vectors from each linear subregion which span the respective manifold segments in that region, not a large or dense sample: The geometry of these $d$-dimensional manifold segments is either an affine subspace, fully specified by a set of $d$ eigenvectors (determined by PCA) plus one support vector, or curved  with the curvature determined by Eq. 4 (exploiting this knowledge about the curvature, in the ideal case we only need $d+1$ support vectors in this case as well, see Appx. D for details, although practically we used kernel-PCA). Hence, the construction in each subregion is analytical, but it is specified through this minimal set of support points.

---

### Official Review · Reviewer_rEg4 · 2025-10-31

**Soundness:** 3
**Presentation:** 2
**Contribution:** 3
**Rating:** 4
**Confidence:** 4

**Summary:**

The authors propose an algorithm to detect stable and unstable manifolds in piecewise linear RNNs. The main challenge is moving across regions, and the algorithm handles this by testing for self-consistency across the boundary, and rejecting inconsistent continuations. The authors use the method to locate boundaries between basins of attraction, and also use it to analyze a chaotic system.

**Strengths:**

Identifying stable and unstable manifolds is an important problem. Most RNN analysis methods deal with fixed points or trajectories.

Limiting the analysis to PLRNN provides a feasible algorithm.

**Weaknesses:**

Discrete vs. continuous time: The definitions in equation 4 (for instance) are of discrete time, where stability is determined by the norm of eigenvalues. But section 3.3 (and later) describe continuous time conditions – negative/positive real part.

RNNs are usually high-dimensional, as mentioned by the authors. Most examples in the paper are low dimensional. And the high-dimensional examples are only visualized in low-D as a check of their validity.

Many results are qualitative, and only visual inspection is provided as a measure of correctness.

The presentation of the paper is not very clear.  For instance, the argument about curved surfaces. If there is a complex eigenvalue pair, it is associated with two eigenvectors spanning a hyperplane. The description in Section 3.3 is of sampling points on this hyperplane. It does not describe following one specific trajectory (which is usually curved also for real eigenvalues).

**Questions:**

Line 133: question mark

The stable manifold is a high-D object in general. Figure 1 is a 1D case.

Line 338: should this be figure 2?

Line 365 – this is a relatively high-dimensional example. But the validation of the reconstruction is only done visually in a 3D projection. It would be useful to quantify the quality of the stable manifold. For instance, running the dynamics forward from nearby points.

Scalability. How does the algorithm scale with system dimensionality? In general, validation on high dimensional examples is not clear.

Figure 5: How does the figure show that the intersections specifically are the chaotic attractor? Figure 5B shows points on all the unstable manifold – not just the intersections.

---

> ### Author Response · Authors · 2025-11-20
> **Rebuttal #1 (Weaknesses)**
>
> We thank the referee for the thorough reading, pointing out several typos and slips, and for the fair assessment of our work!
> **We have uploaded a revised version of our paper with all major changes highlighted in blue.**
>
> **Weaknesses**
>
> **W1 (continuous vs. discrete time):** Thank you for spotting this! Yes, we always meant to refer to the discrete case (as the RNN map  $F_{\\boldsymbol{\theta}}$ is discrete) and are not sure how this slipped in. But as far as we could see, it was only this one occasion (?), which we now fixed.
>
> **W2 (quantitative analysis in high-d):** We now introduce a way of computing exact distances to the manifolds in high-d as a means to arrive at meaningful quantitative conclusions, despite the inherent difficulties in visualizing high-d spaces, see newly added Appx. D. Specifically, in Appx. D we introduce a parameterized form of the manifolds from which we can easily compute distances. We show for the examples of the decision making problem (lines 447-450) and the real neuron (lines 467-470) how this could be used to reveal, for instance, how close the system’s two stable states are to the boundary of the basin of attraction. In the decision making example, for instance, both stable states are about equally far from the basin boundary ($d=0.39$ vs. $d=0.37$), making both choices about equally ‘attractive’. For the real neuron, on the other hand, we find that the limit cycle is relatively vulnerable to perturbations during the spike recovery phase of the oscillation, with the state coming very close to the boundary to the resting state (see sect. 4, lines 467-470). These examples illustrate how we can employ our knowledge about the manifolds also in high-d cases.
>
> **W3 (results qualitative):** We now devised a quantitative measure for assessing the quality of the algorithmically recovered manifolds, based on randomly sampling points on and off the manifold and evaluating their long-term behavior: Points on the un-/stable manifold should converge to a cyclic point in backward, resp. forward, time, hence come arbitrarily close to it, while for points off these manifolds this will not generally be true (they may come close temporarily, but then should shoot off to some other target). We capture this through the following normalized measure
>
> $$
> \delta\_{\sigma}(\\mathbf{x}\_0)
> :=
> \frac{\displaystyle \min\_{k\sigma \ge 0}
> \left\| F\_{\boldsymbol{\theta}}^{k}(\mathbf{x}_0) - \mathbf{p} \right\|_2^{2}}
> {\left\| \mathbf{x}\_0 - \mathbf{p} \right\|_2^{2}}
> \in [0,1].
> $$
>
> where $\\mathbf{p}$ is the saddle under consideration and $\\mathbf{x}_0$ is a point either on the un-/stable manifold or drawn randomly from some neighborhood  $\\mathcal{U}$ ($\sigma=-1$ for the unstable and $\sigma=+1$ for the stable manifold), please see first pg. on p.8 (sect. 4) for further details. Histograms in Appx. Fig. 6 show the distributions of  $\delta\_{\sigma}(\\mathbf{x}_0)$ for points sampled on or off the manifolds for the examples from Figs. 3 \& 4 in the main paper, illustrating that points on the manifold have $\delta\_{\sigma}$ values close to zero, while those sampled off the manifold naturally cover a large range. We accommodate this by defining an index $I\_\mathcal{U}[\delta\_{\sigma}(\\mathbf{x}_0 \in \mathcal{U})>\delta\_{\sigma}^\text{max}] \in \\{0,1\\}$ with $\delta\_{\sigma}^\text{max}:=\max[\delta\_{\sigma}(\\mathbf{x}_0\in W^{\sigma}(\\mathbf{p}) \cap \mathcal{U})]$, based on which we summarize these histograms in the quality statistic
>
>  $$\Delta\_{\sigma}:= \\langle I\_\mathcal{U} \rangle-\\tilde{\delta}_{\sigma}(\\mathbf{x}\_0\in W\^{\sigma}(\\mathbf{p}) \\cap \\mathcal{U}),$$
>
> where $\langle \cdot \rangle$ denotes the mean and $\tilde{\delta}\_{\sigma}$ the median. These values we now report for each of the examples in sect. 4 (see also Table 2 and Fig. 6 in newly added Appx. G), asserting that the manifold reconstructions in all cases were numerically accurate.
>
> **W4 (curved manifolds):** Yes, this is correct of course; complex eigenvalues always come in conjugate pairs, and *within* the specific linear subregion in which the saddle point considered lies, the un-/stable manifolds will always be affine subspaces. This can change, however, as soon as we start entering neighboring subregions, as in the Duffing example in Fig. 4A. The continuation of an un-/stable manifold into a neighboring subregion (as per Eq. 3) may indeed be curved, for instance if the dynamics in that subregion is governed by a true or virtual spiral point (for this is we provided the formulation in Eq. 4, see also Appx. B). And yes, we are considering construction of these manifolds, not of specific trajectories of course which could be curved in any case, as the referee correctly points out.
>
> We have completely rewritten sect. 3.3 and Algorithm 1 (with further edits in sect. 3.2) to make clear how the manifolds are constructed and which properties they have, which should clarify all the points brought up.

---

> > ### Author Response · Authors · 2025-11-20
> > **Rebuttal #2 (Questions)**
> >
> > **Questions**
> >
> > **Q1:** Thanks for pointing out, reference fixed.
> >
> > **Q2:** True, but we found it easier to illustrate how the algorithm works sticking to the 1d case. We made this clear now in sect. 3.3 (line 215) and in the figure legend.
> >
> > **Q3:** Thank you! Yes, fixed.
> >
> > **Q4:** Thank you for this excellent suggestion. As explained in detail in our response to W3 above, we now introduced a quantitative measure for assessing the quality of the manifold reconstruction, along the lines you have suggested.
> >
> > **Q5:** Fig. 9 in Appx. I compared wallclock times for different RNN dimensionalities $M$ and numbers of linear subregions (as given through the number of ReLU nonlinearities $P$). As this graph indicates, within the range tested, the manifold algorithm appeared quite well behaved (we nevertheless also commented on the scaling behavior in the Limitations sect.). For the revision we extended this analysis and evaluated the scaling for different systems, highlighting that for all practical purposes model dimensionality and number of linear subregions don't appear to be limiting factors, see Appx. I and Fig. 9.
> >
> > **Q6:** The intersections in Fig. 5A are only the homoclinic points, not the whole chaotic attractor which happens to lie on the unstable manifold as shown in Fig. 5B. We clarified this in revised sect. 5 and removed the sentence in Fig. 5 legend which may have caused this misunderstanding. Homoclinic intersections inevitably give rise to chaos due to the Smale-Birkhoff Homoclinic Theorem [1,2], an important theorem in dynamical systems theory.
> >
> > **References:**
> >
> > [1] S. Wiggins. Global Bifurcation and Chaos. Springer-Verlag, New York, 1988.
> >
> > [2] Simpson, D. J. (2016). Unfolding homoclinic connections formed by corner intersections in piecewise-smooth maps. Chaos: An Interdisciplinary Journal of Nonlinear Science, 26(7).

---

> > > ### Author Response · Authors · 2025-11-27
> > > **Additional high-d example**
> > >
> > > Further up to our previous response to W2, we have now added another high-dimensional empirical example, human electrocardiograms, to illustrate our algorithm for confirming the presence of chaos through a homoclinic intersection as described in Sect. 5. Chaos is otherwise hard to confirm in this weakly chaotic case, see new Fig. 10 in Appx..

---

> > > > ### Comment · Reviewer_rEg4 · 2025-11-28
> > > > **Quick question**
> > > >
> > > > I apologize for the delay in responding. I read the other reviews and rebuttal and the modified manuscript. Most of my concerns were addressed.
> > > >
> > > > I want to make sure I understand the curved/spiral argument. In the Duffing example - how many subregions are crossed? Can you mark this on Figure 4A? If I understand correctly, within-region is always linear. The spiral you are referring to is a global cross-region spiral, which has to be broken into many subregions. And the complex eigenvalues involved are of the "teacher" system, and not of the linear region-specific systems.
> > > >
> > > > If this is indeed the case, then: 1) It wasn't (at least to me) completely clear from the explanation. No clear distinction of global and local dynamics. 2) In figure 1, steps 3 and 4: should there be more boundaries? 3) Is there some sensitivity in the quality of reconstruction to the curvature of the target system? Intuitively it should demand many more region transitions, and more opportunities for inconsistencies. The chaotic examples are reassuring, but I wonder whether you observed any systematic dependence.

---

> > > > > ### Author Response · Authors · 2025-11-28
> > > > > **Curvature in the manifolds**
> > > > >
> > > > > Thank you for your follow-up question, this is a very important point. To clarify: The manifolds can be curved *even within a single linear subregion*, and hence, to answer **2)**, the representation in Fig. 1 is indeed correct as is, there aren’t any more boundaries. Furthermore, in Fig. 8 in the Appx., we already illustrated the Duffing with linear subregion boundaries drawn in, once again confirming there can be manifold curvature *within* a single linear subregion.
> > > > >
> > > > > However, this aspect is indeed not very intuitive, since - as the referee correctly pointed out - spirals always come with conjugate pairs of complex eigenvalues and hence their un-/stable manifolds always span affine subspaces *within the linear subregion in which the saddle spiral resides*. The crucial point to observe, however, is that a manifold can acquire curvature as soon as it crosses into a neighboring subregion where the dynamics is, for instance, determined by another spiral.
> > > > >
> > > > > To illustrate this, we now created a simple example of a 2d PLRNN, **please see new Fig. 11 in the Appx. (p.30)** of our revision. In this simple example, the stable manifold of the saddle in the first subregion is a straight line. But as soon as it crosses the border into the neighboring subregion where an unstable spiral determines the dynamics, it obtains curvature due to this spiraling dynamics (cf. Eq. 3 for definition of the *global* manifolds). We further emphasized this crucial point on p. 5 (point 3, lines 266-268, sect. 3.3), where we now also refer to Fig. 11. We hope this clarifies the point.

---

### Official Review · Reviewer_KGgx · 2025-11-03

**Soundness:** 3
**Presentation:** 3
**Contribution:** 3
**Rating:** 8
**Confidence:** 4

**Summary:**

The paper proposes an algorithm for detecting stable and unstable manifolds of fixed and cyclic points in piecewise-linear RNNs with ReLU activation functions. The authors claim this is the first algorithm for detecting stable/unstable manifolds in ReLU-based RNNs. The paper then utilizes some dynamical system examples such as Lorenz-63 attractors and single-cell recordings to illustrate the proposed algorithm.

**Strengths:**

- Clarity: detailed math background and derivations, and pseudocodes facilitate the understanding and reproducibility of the proposed approach.

- Mathematically rigorous.

- Validation on across synthetic and real-world problems. And the algorithm handles high-dimensional models with sublinear scaling in subregion sampling. Its efficiency enables previously infeasible analyses, such as identifying attractor basins in decision-making tasks or chaos signatures in electrophysiological data, bridging theoretical insights with empirical applications.

**Weaknesses:**

To begin with, I'd like to disclose that I also reviewed this paper in NeurIPS 2025 and I gave quite positive score. Below I just attach my weaknesses when I reviewed this paper in NeurIPS 2025, and the authors' rebuttal in NeurIPS 2025 have addressed nearly all my concerns.

1. I would have liked a bit more concrete discussion on which parts of the algorithm are heuristic and which are exact. And when they are heuristic, how do potential hyper parameter choices affect the outcome, and how are they picked? I will give some concrete examples here:

- Use of approximate algorithm for finding the fixed points: Is there any concern in case one doesn't find all fixed points?
- Sampling seed points: how many are sampled? How are they sampled? It doesn’t seem trivial (or even always possible) to me to e.g., uniformly sample within one region of linear dynamics.
- Propagating seed points. I think one is not always sure we reach all reachable manifolds from a given sub-region? (E.g., if we undersample seed points, or all seed points / most of the space ends up converging in one particular direction).
- The fallback algorithm 3 still doesn’t have any guarantees of finding the right subspace, does it?

2. From results shown it is unclear to me that the regularisation (Eq. 6) actually reaches the desired effect. Besides the effect on training, does it actually lead to networks that are more likely to be invertible? Is there some way to verify this, maybe by comparing the average condition numbers of the to be inverted matrices when running your manifold construction algorithm on a model trained with and without the regulariser?

**Questions:**

See Weakness

---

> ### Author Response · Authors · 2025-11-20
> **Rebuttal**
>
> Thank you very much for disclosing you have been a referee for our previous NeurIPS submission, and for all your support and positive assessment of our work (since all referees had voted for acceptance on that previous submission, we were taken a bit by surprise it nevertheless ended up as a rejection and hadn’t implemented all changes yet – but we have done now!). We briefly recapitulate our previous responses and the changes we have now implemented. **All text changes are highlighted in blue in the revised manuscript.**
>
> **Weaknesses**
>
> **W1 (heuristic vs. exact parts of algo):** We have completely rewritten sect. 3.3, which introduces the core algorithm, and the pseudo-code for Algorithm 1. We hope that all questions are now answered in this rewrite. Briefly: *Within* any given linear subregion, we have in principle an analytical construction for the respective manifold segment which does not depend on any hyper-parameters. Numerical issues, however, can arise in cases of strong anisotropy in the vector field, so we took steps to ensure that slow eigendirections are properly represented within the set of support points needed to anchor a piece of manifold. The algorithm’s main numerical (“heuristic”) aspect is finding the extensions of the manifolds across boundaries into neighboring subregions, and the only real hyper-parameter ($N_s$, the number of sample points to probe extensions into neighboring regions) is associated with this process.
>
> On your detailed points:
>
> - Please note that the algorithm for finding fixed and cyclic points (SCYFI [1]) itself is, strictly, not part of the present algorithm. It is approximate in the sense that not all fixed points may be found, but the fixed points themselves will be exact. While SCYFI has been shown to have generally good convergence properties [1], if not all fixed points are found this simply means that we cannot determine all of the un-/stable manifolds. Whether this is of concern we would argue depends more on the specific application case, as often obtaining a local picture of the state space may be sufficient. But this may be important to point out as a potential limitation, and so we added this now to our discussion.
>
> - First, for clarity, the $N_s$ seed points are not sampled across a whole linear subregion, but only from the un-/stable manifold for which we are trying to determine the extension into neighboring subregions (whether we draw these points from a uniform or a Gaussian distribution, for instance, is not crucial, nor do we need to cover the whole manifold). Second, note that to uniquely position a segment of the $d$-dimensional manifold in a given linear subregion, $d+1$ support points are usually* sufficient, which are deterministically generated by forward (resp. backward) propagating with the RNN map $F_{\theta}$ (*: this is strictly true if the manifolds are affine subspaces, but also if they are curved and we can use our knowledge about the type of curvature expressed in Eq. 4, see new Appx. D). We then only need to make sure that we reach the continuations of the manifold into all connecting subregions. Say we have $k_{neigh}$ neighboring subregions, a good heuristic is simply $N_s = k_{neigh}  \times d$ (as pointed out now).
>
> - This is, in principle, correct, we cannot strictly guarantee all linear subregions are reached (once a subregion is reached, however, we can guarantee reconstruction of the full manifold within that subregion). To make sure, one could in principle increase $N_s$ until the number of detected continuations of the manifold into other subregions does not increase anymore (we have added this as a footnote comment on p.5). In the case of the Duffing system, for instance, we were able to reconstruct the manifold with just 4 sample points per region.
>
> - Here the same argument applies as above, i.e. we could in principle examine the dependence on $N_s$ (the number of sample points) to determine the point where the number of accessed subregions plateaus.
>
> **W2 (effect of regularisation):** Yes, the regularisation does produce the desired effect, as shown in Fig. 2A (top-left). This was a bit buried in a half-sentence, but we hope it is now much clearer in sect. 3.4 (below Eq. 5) in our rephrasing and reordering of the points. Please also note that by ‘invertibility’ we do not mean invertibility of the model’s Jacobian (transition matrix), but we mean invertibility in the sense defined on line 304 of the manuscript (i.e., such that the mapping is bijective across boundaries). For this we need the determinants of all Jacobians to have the same sign (Fujisawa et al. 1972), as encouraged by Eq. 5. We need this condition for backtracking the stable manifolds across linear subregions. We tried to emphasize this in the revision in the first sentence of sect. 3.4.

---

### Author Response · Authors · 2025-11-27
**Any comments on our revisions?**

Dear Referees,

It’s now been a week since we have uploaded our revised manuscript and rebuttal. Since the discussion deadline (and the NeurIPS conference!) is quickly approaching, we wondered whether you would have any feedback on the extensive revisions we have done? Thank you!

The Authors

---

### Public Comment · ~Matthijs_Pals1 · 2025-11-28
**Reviewer KGgx's review**

While I personally agree with their score, I can confidently assess that Reviewer KGgx **did not write** the NeurIPS review they copied over (I hope they just accidentally copied over the wrong review).

Edit: I had not previously realised that my comment might negatively affect the review process (as KGgx gave the highest score), which absolutely was not my intention, and I hope won't be the case. I do want to state that irrespective of a potential copy error in KGgx's review, indeed *all* NeurIPS 2025 reviewers supported this paper after the review process there.

---

### Author Response · Authors · 2025-12-01
**Final summary**

Dear Area Chair,

We are herewith taking up the suggestion of the program chairs to provide a final summary of our revisions and the rebuttal process:

The two referees with the highest scores (KgUd: 6, KGgx: 8) were quite enthusiastic and only requested minor clarifications in the text or discussion of additional literature, which we all implemented.

One referee (rEg4, initial score 4) also stated that we addressed an important problem, but requested more extensive changes, most importantly a quantification of the qualitative results in the paper. This we have all provided, including a new quality measure and a procedure for computing metrics on parameterized manifolds in high-dimensional spaces. In their last response before the “lock-down” the referee thus indicated *that their concerns were mostly addressed*, except one further clarification on the question of curvature of the manifolds. Here there was a misunderstanding, namely that within a given subregion manifold segments can only be affine subspaces (resulting in some follow-up questions based on this wrong assumption). Thus, in our last response, we clarified by further textual changes (Sect. 3.3) that manifold segments can indeed also be curved *within* a single subregion, and created a concrete example illustrating how such curvature arises (Fig. 11 in Appx.). With this final clarification we would therefore assume this referee would have been satisfied with our final revision.

Finally, referee MbXp (initial score 4) stated that they found the paper relevant, interesting, and well written and presented, *addressing a very hard problem*, with our paper being one of the first to provide a solution for these systems in high-d. Their main issues were just textual in nature, namely that many algorithmic details were not clear enough, while in other places there were too many details and the Appx. was too unstructured and extensive. These points we could see, and have thus rewritten the core algorithmic Sect. 3.3 all from scratch to improve clarity, have completely overhauled the presentation of main Algorithm 1 itself, have removed excessive detail in other places, and have completely reworked the whole Appx.. We thus think we have thoroughly addressed and fixed all their weaknesses and questions.

To **summarize the changes and revisions we made** (all textual changes marked in blue in the revised paper):

- Designed a new quantitative measure (Eq.6, lines 381-395, p.8, new Appx. G and Fig. 6) to assess the quality of the manifold reconstructions, used for all examples in the subsequent pg.s. (rEg4)

- Designed a new parameterization of the curved global high-dimensional manifolds that enables easy computation of metrics (new Appx. D), which we used throughout sects. 4 \& 5 to provide further quantitative results (blue text additions in these sect.). (rEg4)

- Added another high-dimensional empirical (medical) example, human ECG, on which we illustrated our algorithm in a real-world setting (new Fig. 10 and corresponding text changes). (rEg4)

- Added another test on the scaling behavior of our algorithm, new Fig. 9C. (rEg4)

- Added an illustrative example explaining how curvature in the manifolds can arise in single subregions (new Fig. 11). (rEg4, KGgx)

- Completely rewrote core algorithmic sect. 3.3 and Algorithm 1, clarifying open points and adding all relevant mathematical details. (MbXp, KGgx)

- Completely restructured the Appx., rewrote and clarified larger parts of it, removed parts not directly relevant to the main paper, and added further details where they were requested (blue text in Appx. B, C, E, H, and I). (MbXp)

- Added discussion of further literature, as pointed out by referees KgUd \& MbXp, to Introduction (p.2, last pg.) and Conclusions (p.10, 2nd pg.).

- Made a number of other textual changes to clarify a variety of minor points (effect of regularization, discrete vs. continuous dynamics, usage of SCYFI etc.).

Our paper is indeed the first, to our (and Perplexity’s) knowledge, that provides a semi-analytical construction of global unstable and stable manifolds of fixed points and cycles in RNNs that works even in high dimensions. These manifolds are crucial to understand and dissect the dynamical behavior of RNNs, and thus to gain insight into how they solve tasks and reconstruct the dynamics of empirically observed systems. This, in turn, is important for explainable AI and for applications in science and medical areas where mechanistic insight into system dynamics is sought.

Thank you for considering these points,

The Authors

---

### Meta-Review · Area_Chair_ubeL · 2026-01-05

**Summary:**

The paper introduces an algorithm to compute stable and unstable manifolds of fixed points and periodic points in piecewise-linear RNNs (PLRNNs) with ReLU activations. These manifolds are used to (i) delineate basins of attraction (multistability) and (ii) detect homoclinic intersections whose existence implies chaos. Many numerical examples are provided to support the effectiveness of the proposed algorithm.

**Reviewer Concerns:**

The initial reviews of the paper feature a mix of positive and negative assessments (Reviewer KGgx and KgUd are positive; Reviewer rEg4 and MbXp are borderline negative). All the reviewers recognize the importance of the problem studied in the paper and appreciate the nontrivial attempt made the authors for addressing a challenge problem. The writing of the paper is considered relatively good. Nonetheless, there are concerns raised by the reviewers from different perspectives:

1) whether the regularization achieves intended invertibility-related effect and how to verify it (Reviewer KGgx and MbXp);

2) confusing discrete-time vs continuous-time stability language (Reviewer rEg4);

3) algorithm clarity and structure (Reviewer MbXp and KGgx);

4) Insufficient empirical evaluation metrics (Reviewer rEg4).

**Reviewer Scores:**

The authors provide detailed responses to concerns raised by the reviewers, and the manuscript is carefully revised. Major revisions are

1) a clearer description of an invertibility “sign condition” and a regularization term added to the training loss (Eq. 5), plus empirical plots describing invertibility impacts in Section 3.4;

2) a fix to the discrete/continuous mismatch—discrete-time terms are used throughout;

3) a rewrite to Section 3.3 and Algorithm 1 from scratch;

4) a new quantitative quality measure for manifold reconstruction (Eq. 6; plus new appendix material and summary tables/figures).

These revisions directly target at the concerns from reviewers and Reviewer rEg4 posted a post-rebuttal response commenting that most concerns were resolved and there were several new clarification questions. Given the detailed revisions and responses, Reviewer rEg4 is likely to increase his/her initial rating. The concern from Reviewer MbXp is also likely to be explained.

---

### Decision · Program_Chairs · 2026-01-26

Accept (Poster)